# Genetic and functional diversity of β-*N*-acetylgalactosamine-targeting glycosidases expanded by deep-sea metagenome analysis

Tomomi Sumida ●[1] ✉, Satoshi Hiraoka ●[1], Keiko Usui ●[1], Akihiro Ishiwata ●[2], Toru Sengoku ●[3], Keith A. Stubbs ●[4], Katsunori Tanaka[2,5], Shigeru Deguchi ●[1], Shinya Fushinobu ●[6] ✉ & Takuro Nunoura ●[1]

β-*N*-Acetylgalactosamine-containing glycans play essential roles in several biological processes, including cell adhesion, signal transduction, and immune responses. β-*N*-Acetylgalactosaminidases hydrolyze β-*N*-acetylgalactosamine linkages of various glycoconjugates. However, their biological significance remains ambiguous, primarily because only one type of enzyme, exo-β-*N*-acetylgalactosaminidases that specifically act on β-*N*-acetylgalactosamine residues, has been documented to date. In this study, we identify four groups distributed among all three domains of life and characterize eight β-*N*-acetylgalactosaminidases and β-*N*-acetylhexosaminidase through sequence-based screening of deep-sea metagenomes and subsequent searching of public protein databases. Despite low sequence similarity, the crystal structures of these enzymes demonstrate that all enzymes share a prototype structure and have diversified their substrate specificities (oligosaccharide-releasing, oligosaccharide/monosaccharide-releasing, and monosaccharide-releasing) through the accumulation of mutations and insertional amino acid sequences. The diverse β-*N*-acetylgalactosaminidases reported in this study could facilitate the comprehension of their structures and functions and present evolutionary pathways for expanding their substrate specificity.

Beta-*N*-acetylgalactosamine (β-GalNAc)-containing glycans, such as glycoconjugates of glycolipids[1,2], polysaccharides[3–5], *N*- and *O*-linked glycans[6,7], O-antigen[8] and glycosaminoglycan (chondroitin sulfate)[9] (Supplementary Fig. 1), are ubiquitous and crucially contribute to various biological processes, including cell adhesion, signal transduction, cross-interactions with functional membrane components, formation of the cell envelope and maintenance of its stability, immunomodulation, and immune responses[1–9]. The regulatory function of these glycans is attributed to their structural diversity, which

differs in carbohydrate constituents (namely, glucose, galactose) and molecular architecture (α- or β-linkages, linear or branched)[1–10].

Beta-*N*-acetylgalactosaminidases (β-NGAs) hydrolyze the different β-GalNAc linkages of various glycans to modulate the length, combination, and abundance of glycans. This catalytic activity requires the β-NGAs to possess diverse substrate specificities, although only two enzymes, exo-β-NGA and exo-β-*N*-acetylhexosaminidase (exo-β-HEX), possessing distinctive substrate specificity and sequence, have been demonstrated to hydrolyze β-GalNAc. Exo-β-NGAs have a strict

¹Research Center for Bioscience and Nanoscience, Japan Agency for Marine-Earth Science and Technology (JAMSTEC), Yokosuka, Japan. ²RIKEN, Cluster for Pioneering Research, Wako, Saitama, Japan. ³Department of Biochemistry, Yokohama City University Graduate School of Medicine, Kanazawa-ku, Yokohama, Japan. ⁴School of Molecular Sciences, The University of Western Australia, Crawley, WA, Australia. ⁵Department of Chemical Science and Engineering, Tokyo Institute of Technology, Meguro-ku, Tokyo, Japan. ⁶Graduate School of Agricultural and Life Sciences, The University of Tokyo, Bunkyo-ku, Tokyo, Japan. ✉e-mail: sumidat@jamstec.go.jp; asfushi@mail.ecc.u-tokyo.ac.jp

substrate specificity for the non-reducing terminal β-GalNAc and are classified into the glycoside hydrolase (GH) family GH123[11] of the Carbohydrate-Active Enzymes (CAZy) database[12,13]. Previously reported exo-β-NGAs have been identified in only three bacterial species, specifically associated with microbe-host interactions in both terrestrial soil and human gut environments (namely, NgaP from *Paenibacillus* sp. TS12, CpNga123 from *Clostridium perfringens*, and BvGH123 from *Phocaeicola vulgatus*)[11,14,15], with no reported origin in archaea or eukaryotes, and endo-β-NGA has not been reported. In addition to these three species, Moreno-Prieto et al. reported a β-NGA (NaNga) from the terrestrial soil bacterium *Niabella aurantiaca* during the revision of this manuscript[16]. Meanwhile, exo-β-HEXs hydrolyze the non-reducing terminal of β-GalNAc as well as β-*N*-acetylglucosamine (GlcNAc) and are mainly classified into the family GH20[17–19]. More than 100 exo-β-HEXs from all three domains of life have been functionally analyzed, with no report of endo-β-HEXs.

Given the limited extant knowledge on β-NGAs, it is imperative to further identify and functionally characterize β-NGAs. These endeavors are critical for comprehensively understanding the complex phenomena associated with β-GalNAc-mediated biological processes. Recently, culture-independent metagenomic exploration of glycosidases has substantially augmented our conception of carbohydrate-related enzymes. Function-based screening of diverse biological resources revealed several glycosidase families, including GH148 from volcanic soil[20], GH156 from a thermal hot spring[21], GH165 from agricultural soil[22], and GH173 and CBM89 from the capybara intestine[23]. Moreover, sequence-based screening has enabled the analysis of much larger metagenomic sequencing datasets than functional screening, yielding more candidate sequences and facilitating the discovery of enzymes with distinct characteristics[24].

Beta-*N*-acetylgalactosamine is prevalent in various glycans across the three domains of life (Bacteria, Archaea, and Eukarya) in different ecological niches[1–10], and β-NGA likely follows a similar distribution pattern. However, β-NGAs have only been identified from four bacterial species of Bacillota and Bacteroidota in terrestrial soil and human gut environments, and their use has been discussed only in limited environments and species. Thus, we searched for β-NGAs using a sequence-based screening approach against environmental metagenomes. Particularly, we focused on the deep-sea environments to investigate the role of β-GalNAc in natural biological processes. The deep sea is characterized by unique features and distinct bacterial flora that are quite different from terrestrial environments[25] but is underexplored owing to the challenges associated with sampling from these regions, making the deep-sea metagenomes a promising frontier for the discovery of enzymes[26,27]. Deep-sea metagenomes hold potential for the discovery of β-NGA genes, considering the prevalence of β-GalNAc in bacterial and archaeal exopolysaccharides[4–6] and chondroitin sulfate[28] in marine environments.

In this work, our exploration against deep-sea sediment metagenomes (DSSM) and following domain search using a public gene sequence database yielded four β-NGA gene subfamilies that are phylogenetically distinct from known GH123 exo-β-NGAs. The biochemical and structural characterization of these enzymes not only unveiled their functional diversity but also shed light on their monophyletic evolutionary history, providing valuable insights into the mechanisms underlying β-GalNAc-mediated biological processes.

## Results

### Discovery of a β-NGA from deep-sea sediment metagenomes
Using four metagenomic datasets derived from microbial assemblages in deep-sea abyssal sediments and a domain-based search, we retrieved three candidate complete coding sequences (CDSs) (Gene ID: *dssm_1–3*, tentative Protein ID: DSSM_1–3), which exhibited low sequence identity (15–26%) to all known GH123 exo-β-NGA genes (Protein ID: NgaP, CpNga123, and BvGH123) (Supplementary Fig. 2a,

Supplementary Data 1). Sequence alignments demonstrated that the consecutive catalytic DE motif of family GH123, comprising an aspartic acid (stabilizer of the 2-acetamido group of the substrate) and a glutamic acid (acid/base)[11], was present in the *dssm_2* and *dssm_3* sequences but not in *dssm_1* (Supplementary Fig. 2b, green box and asterisk).

AlphaFold2[29] was used to predict the structure of the candidate proteins (Fig. 1a). DSSM_1 was structurally distinct from the GH123 β-NGAs (Fig. 1a, b) and consisted of five domains, among which only domain 2 (β-sandwich) displayed a degree of structural similarity to the N-terminal domain of GH123 β-NGAs. Domain 4 ((β/α)$_8$ barrel) was similar to GH66 cycloisomaltooligosaccharide glucanotransferase (PDB, 3WNM), although a blast search using the *dssm_1* sequence did not find significant similarity with enzymes registered in PDB. Based on the structural feature, *dssm_1* was not considered to be a β-NGA candidate sequence. In contrast, the predicted structures of DSSM_2 and DSSM_3 were similar to those of GH123 β-NGAs, and they all shared DUF4091, a presumed domain region whose function remains undetermined (Fig. 1b, green, Supplementary Fig. 2b, green box). No sequence corresponding to DUF4091 was conserved in *dssm_1*, although alignment of the sequences corresponding to DUF4091 in *dssm_2*, *dssm_3* and the previously reported GH123 β-NGAs was confirmed and showed higher identities than the full-length sequences (Supplementary Fig. 2c) and similar amino acid conservation as the HMM logo in the Pfam database (Supplementary Fig. 2d, e).

Subsequently, the β-NGA candidate sequences *dssm_2* and *dssm_3* were selected and heterologously expressed in *Escherichia coli*. The encoding sequences lacking the predicted signal peptides (Supplementary Data 1, bold and underlined) were cloned into an expression vector. Although an expression construct of *dssm_2* failed to yield a soluble protein, the enzyme encoded by *dssm_3* was solubilized, and a purified protein was successfully obtained (renamed Protein ID: NgaDssm). Assays using various *p*NP-substrates revealed that NgaDssm was active on GalNAc-β-*p*NP but not on GlcNAc-β-*p*NP or GalNAc-α-*p*NP, indicating that the enzyme possessed exo-β-NGA but not exo-β-HEX activity. Furthermore, the enzyme hydrolyzed Galβ1-3GalNAc-β-*p*NP but not Gal-β-*p*NP, demonstrating a oligosaccharide (including disaccharide)-releasing (OR)-β-NGA activity in addition to monosaccharide-releasing (MR)-type (exo-type) activity (Table 1). These findings suggest that NgaDssm is an OR/MR-β-NGA. Hereinafter, exo-type is tentatively referred to as MR-type in this paper.

### Phylogenetic diversity of β-NGAs
The uncharacterized domain DUF4091 was conserved in the GH123 MR-β-NGAs and an OR/MR-β-NGA gene sequence. Consequently, we utilized DUF4091 as a query to further identify β-NGA genes. We retrieved 734 sequences containing DUF4091 from the Pfam protein family database. The catalytic DE motif was highly conserved (94%), and thus, the majority of these genes were expected to encode enzymes possessing β-NGA activity. A phylogenetic tree of these sequences, along with the three known GH123 β-NGA genes and the two deep-sea β-NGA candidates (*dssm_2* and *dssm_3*) (Supplementary Data 2), identified five major groups: GH123, Group 1, Group 2, Group 3 and Group 4 (Fig. 1c). The *dssm_3* (NgaDssm) has belonged to Group 3 while *dssm_2* was assigned to Group 1 (Fig. 1c).

All previously reported GH123 enzymes were exclusively derived from specific bacteria lineages (i.e., *Bacillota* (NgaP and CpNga123) and *Bacteroidota* (BvGH123)). In this study, to comprehensively examine diverse β-NGAs in nature and enhance our understanding of the enzyme, we manually selected 14 sequences from a range of diverse organisms (plants, archaea, and various bacterial phyla) for thorough coverage of the four Groups defined based on phylogeny (Fig. 1c): for Group 1, *Cohnella abietic* from *Bacillota* (NgaCa), *Meiothermus granaticius* from *Deinococcota* (NgaMg), *Nostoc punctiforme* (NgaNp), *Clyndrospermum stagnale* (NgaCs), and *Stanieria cyanosphaera* (NgaSc)

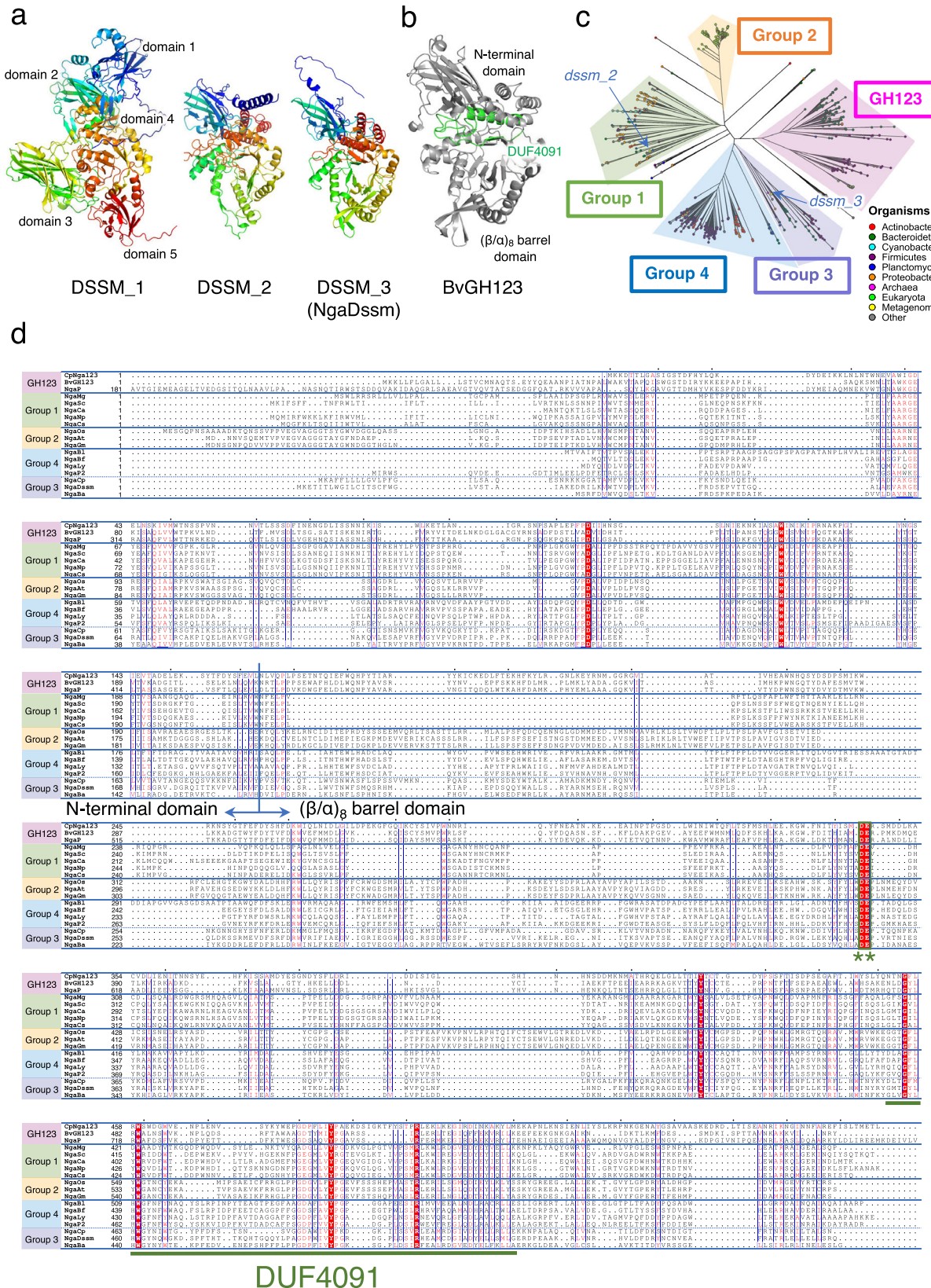

**Fig. 1 | Candidate β-NGA sequences retrieved from deep-sea sediment metagenomes and phylogenetic and sequence diversities of β-NGA candidates.** **a** Overall structures of β-NGA candidates retrieved from deep-sea sediment metagenomes. The structures were predicted using AlphaFold2. **b** The structure of BvGH123. DUF4091 is colored in green. **c** Phylogenetic tree of β-NGA candidates retrieved from deep-sea sediment metagenomes and the Pfam database. Sequences used for the phylogenetic tree of β-NGA candidates were listed in Supplementary Data 2. **d** Alignment of the β-NGA gene sequences. Residues conserved between all the analyzed proteins are shown in a red background. The conserved DE residues (green asterisk, *) are indicated by a green box. The DUF4091 region is underlined.

**Table 1 | Substrate specificities and general properties of β-NGAs**

| | NgaDssm Group 3 | NgaCa Group 1 | NgaMg Group 1 | NgaAt Group 2 | NgaBa Group 3 | NgaBf Group 4 | NgaBl Group 4 | NgaLy Group 4 | NgaP2 Group 4 | NgaP[a] GH123 |
|---|---|---|---|---|---|---|---|---|---|---|
| **Substrate specificity** | | | | | | | | | | |
| **Substrate** | **Relative activity (%)** | | | | | | | | | |
| GalNAc-β-pNP | 100 | – | – | 100 | 100 | 100 | 100 | 100 | 100 | 100 |
| Galβ1-3GalNAc-β-pNP | 70.3 | 100 | – | – | 7.3 | – | – | – | – | NT |
| Gal-β-pNP | – | – | – | – | – | – | – | – | – | – |
| GlcNAc-β-pNP | – | – | – | 13.6 | – | – | – | – | 4.7 | – |
| GalNAc-α-pNP | – | – | – | – | – | – | – | – | – | – |
| GalNAc-β-4MU | 100 | – | – | 100 | 100 | 100 | 100 | 100 | 100 | 100 |
| GalNAc4S-β-4MU | 183.8 | – | – | 47.9 | 1.5 | 1.9 | 2.0 | 1.3 | 29.0 | – |
| GalNAc6S-β-4MU | – | – | – | – | – | – | – | – | – | – |
| **General property** | | | | | | | | | | |
| **Enzyme activity** | **OR/MR-β-NGA** | **OR-β-NGA** | **Unknown** | **OR/MR-β-HEX** | **OR/MR-β-NGA** | **MR-β-NGA** | **MR-β-NGA** | **MR-β-NGA** | **MR-β-NGA** | **MR-β-NGA** |
| Optimal pH | 5.0 | 5.0 | – | 5.0 | 6.5 | 6.5 | 5.0 | 7.0 | 5.5 | 6.0 |
| Optimal buffer | Citrate | Citrate | – | Citrate | Citrate | Citrate | Citrate | HEPES | Citrate | Acetate |
| Optimal temperature (°C) | 45 | 25 | – | 40 | 70 | 35 | 45 | 40 | 40 | NT |
| $T_m$ (°C) | 63.9 | 43.1 | 70.6 | 52.9 | 93.1 | 44.3 | 57.2 | 59.7 | 56.8 | NT |
| $K_m$ (mM) | (3.8 ± 0.4) | (11 ± 3) | – | (12 ± 5) | (3.9 ± 1) | (1.0 ± 0.07) | (2.8 ± 0.4) | (1.0 ± 0.05) | (0.36 ± 0.03) | 0.4 |
| $k_{cat}$ (s⁻¹) | (43 ± 3) | (1.2 ± 0.3) | – | (8.0 ± 3) | (510 ± 50) | (170 ± 5) | (390 ± 40) | (480 ± 10) | (28 ± 0.6) | 7.3 |
| $k_{cat}/K_m$ (s⁻¹ mM⁻¹) | (11) | (0.11) | – | (0.64) | (130) | (170) | (140) | (490) | (79) | 21 |

–, activity < 0.5%. Values are presented as means of technical triplicate experiments. The general properties of each β-NGA are listed based on the results in Extended Data Fig. 3. Controls were measured without the enzyme, serving as blanks, and the relative activity was calculated as 100% of the activity of GalNAc-β-pNP (or Galβ1-3GalNAc-β-pNP for NgaCa) or GalNAc-β-4MU. The enzymes that did not reach saturation are indicated in parentheses in Table 1 owing to their apparent values.

*NT* not tested.

[a]NgaP values are listed based on previous reports[11,32].

from *Cyanobacteria*; for Group 2, *Arabidopsis thaliana* (NgaAt), *Glycine max* (NgaGm), and *Oryza sativa* (NgaOs) from plants; for Group 3; *Candidatus Bathyarchaeia archaeon* B24 from *Thermoproteota* in Archaea from the hydrothermal vent microbiome (NgaBa) and *Chitinophaga pinensis* from *Bacteroidota* (NgaCp); for Group 4; *Brachybacterium faecium* (NgaBf) and *Bifidobacterium longum* subsp. *infantis* (NgaBl) from *Actinomycetota*, *Lacticaseibacillus yichunensis* (NgaLy), and *Paenibacillus* sp. TS12 (NgaP2) from *Bacillota* (Supplementary Data 2 and 3). Overall, the sequence identities between each group of 14 candidate genes and the three GH123 genes were low (12–26%) based on sequence alignment (Supplementary Fig. 3a), and only nine amino acids were entirely conserved (Fig. 1d, red background). DUF4091 was a comparatively well-conserved region, comprising four of the nine strictly conserved amino acids, with its sequence identity being at least 10% higher than that of the full-length sequence (Supplementary Fig. 3b, c).

## Substrate specificity of β-NGA candidates

Recombinant expression vectors for the candidate genes (except for NgaGm and NgaOs) were constructed to assess the activity of the identified enzymes (Supplementary Data 3). Although four expression constructs (NgaNp, NgaCs, NgaSc, and NgaCp) failed to yield soluble proteins, the remaining eight (NgaCa, NgaMg, NgaAt, NgaBa, NgaBf, NgaBl, NgaLy, and NgaP2) successfully expressed soluble enzymes (Supplementary Fig. 3a, red letter) and were subjected to protein purification for subsequent enzymatic assays (Fig. 2a, Table 1).

The substrate specificity of the expressed enzymes was assayed using various synthetic substrates (Table 1). Surprisingly, Group 1 NgaCa displayed strictly OR-type enzymatic activity and acted solely on Galβ1-3GalNAc-β-pNP. The other Group 1 protein, NgaMg, demonstrated no activity against any of the tested substrates.

In Group 2, NgaAt was active against GalNAc-β-pNP and GlcNAc-β-pNP but not against Galβ1-3GalNAc-β-pNP, indicating that it possessed MR-β-HEX activity. However, NgaAt did not share notable sequence

similarity with any of the three types of β-HEXs (HEXO1–3) classified into the GH20 family from *A. thaliana* (Supplementary Fig. 4a, b)[30,31]. Thus, NgaAt is a different lineage of β-HEX from GH20.

NgaBa in Group 3 was active on both GalNAc-β-pNP and Galβ1-3GalNAc-β-pNP, similar to NgaDssm, suggesting that members of Group 3 share OR/MR-type β-NGA activity. In Group 4, NgaBf, NgaBl, and NgaLy were only active on GalNAc-β-pNP, highlighting their strict MR-β-NGA functionality. NgaP2 was active against GalNAc-β-pNP and had weak activity against GlcNAc-β-pNP. Collectively, these results showed that the enzymes in each group exhibit a characteristic OR- and/or MR-type cleavage mode and substrate specificity.

NgaP from GH123 is an excellent tool for detecting sulfatase deficiency, as it does not act on substrates sulfated at positions 4 or 6 of GalNAc[32] and has been successfully used for screening of mucopolysaccharidosis (a metabolic disorder caused by the accumulation of mucopolysaccharides) in newborns[32,33]. We evaluated whether the enzymes identified herein possessed GalNAc4S or GalNAc6S cleavage activity. Using GalNAc4S-β-4MU and GalNAc6S-β-4MU (Table 1), we observed that NgaDssm, NgaAt, and NgaP2 acted on GalNAc4S-β-4MU. In particular, NgaDssm displayed approximately two-fold higher activity against GalNAc4S-β-4MU than against GalNAc-β-4MU.

## Generic properties of the β-NGAs

Next, we evaluated the enzyme characteristics (Table 1, Fig. 2). The optimal pH for the eight enzymes was in the range of 5.0–7.0. NgaBa depicted a particularly broad optimal pH range, with relative activity above 90% at pH 5.5–7.5. Metal ions generally did not affect enzyme activity, except that of NgaLy, which was inhibited by Ni²⁺, Co²⁺, and Zn²⁺ (Table 1, Fig. 2b, c). Overall, the enzymes exhibited moderate temperature optima (25–45 °C) except for NgaBa, an enzyme from a hydrothermal vent microbiome[34], which displayed a high optimal temperature (70 °C) and $T_m$ value (93.1 °C). Thus, NgaBa is the thermostable β-NGA reported to date. The melt curve plot suggested that

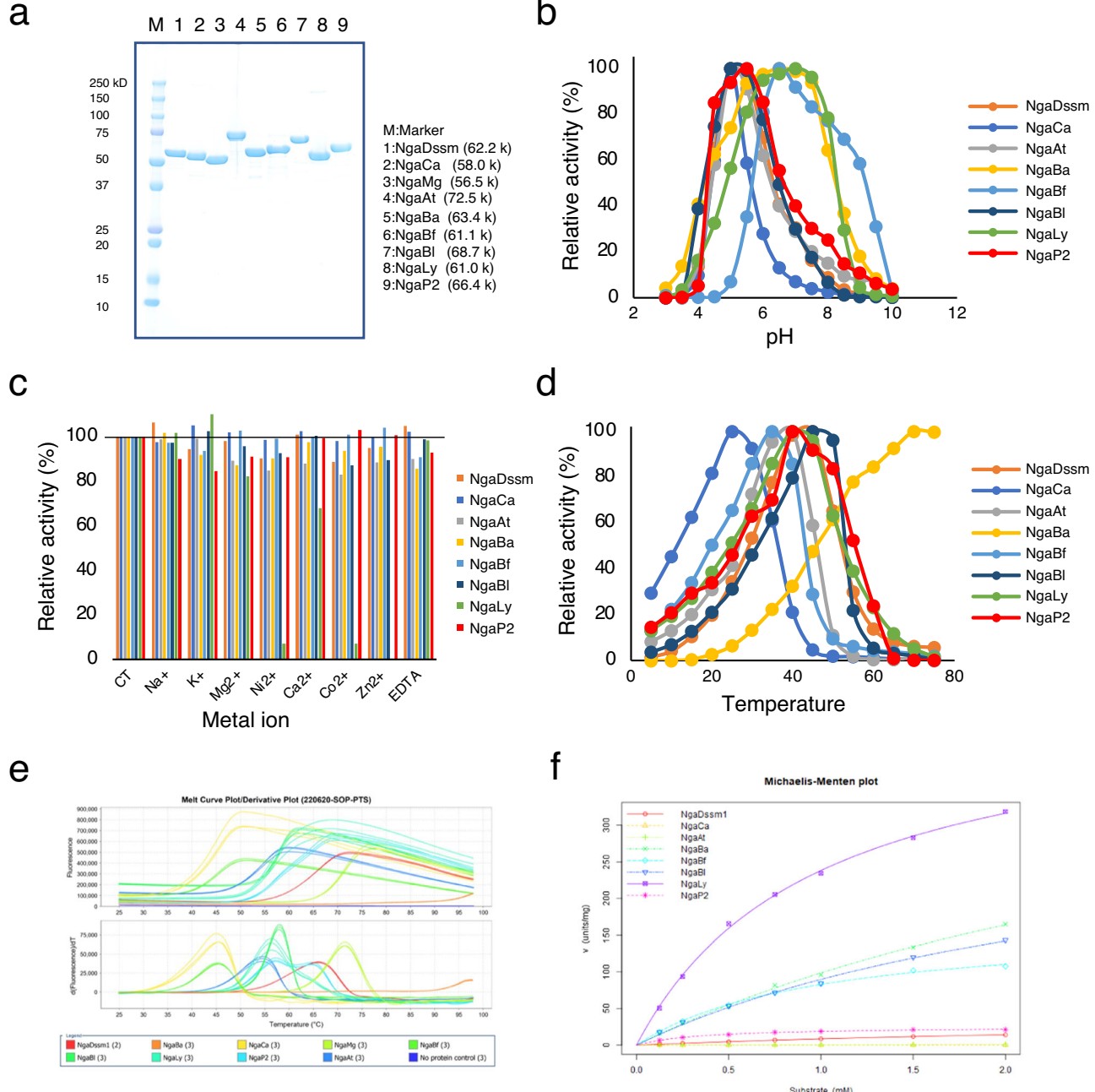

**Fig. 2 | General properties of recombinant β-NGAs. a** SDS-PAGE analysis of recombinant β-NGAs, stained with Coomassie brilliant blue. Effect of (**b**) pH and (**c**) metal ions on the enzymatic activity of recombinant β-NGAs. **d** Effect of temperature on enzymatic activity of recombinant β-NGAs. **e** Protein thermal shift data of recombinant β-NGAs. **f** s-v plots of recombinant β-NGAs. All values represent the mean of triplicate measurements.

the optimal temperature was very close to the thermal denaturation point, with the optimal temperature being approximately 10–20 °C lower than the $T_m$ value (Fig. 2d, e). The $K_m$ and $k_{cat}$ of each enzyme for the most preferred substrate (GalNAc-β-$p$NP or Galβ1-3GalNAc-β-$p$NP) were also examined (Fig. 2f), wherein NgaLy possessed the highest $k_{cat}/K_m$ value. Kinetic assays were performed at a maximum substrate concentration of 2 mM as GalNAc-β-$p$NP was insoluble at buffer concentrations exceeding 3 mM. Nevertheless, all the assayed enzymes did not attain saturation under the aforementioned conditions (Fig. 2f, Supplementary Fig. 5). Therefore, these enzymes exhibited low affinities for $p$NP-substrates; these kinetic quantities were deemed apparent values owing to the failure to accomplish saturation. Table 1 presents the aforementioned apparent values within parentheses.

## Substrate specificity for oligosaccharides

The substrate specificity of each enzyme was explored using various oligosaccharides as substrates (Fig. 3a, Supplementary Table 1, Supplementary Fig. 1, glycolipid oligosaccharide; ganglio-series and globo-series). NgaCa and NgaMg from Group 1 and NgaDssm from Group 3 did not act on GA1 and Gb5 oligosaccharides, although these two oligosaccharides shared the non-reducing end structure Galβ1-3GalNAc-β- with Galβ1-3GalNAc-β-$p$NP (Fig. 3b, c, e). NgaAt acted on GalNAcβ1-3Gal- but not on GalNAcβ1-4Gal- (Fig. 3d). In contrast, NgaBa was functional against GA1 and Gb5 (Fig. 3f and Supplementary Fig. 6b, d, lanes 2 and 5) and degraded GA1 better than GA2 (Fig. 3f and Supplementary Fig. 6b, d, lanes 2 and 4). The OR-activity of NgaBa against β-GalNAc located inside the oligosaccharide was stronger than its MR-activity against β-GalNAc at the non-reducing ends, although this

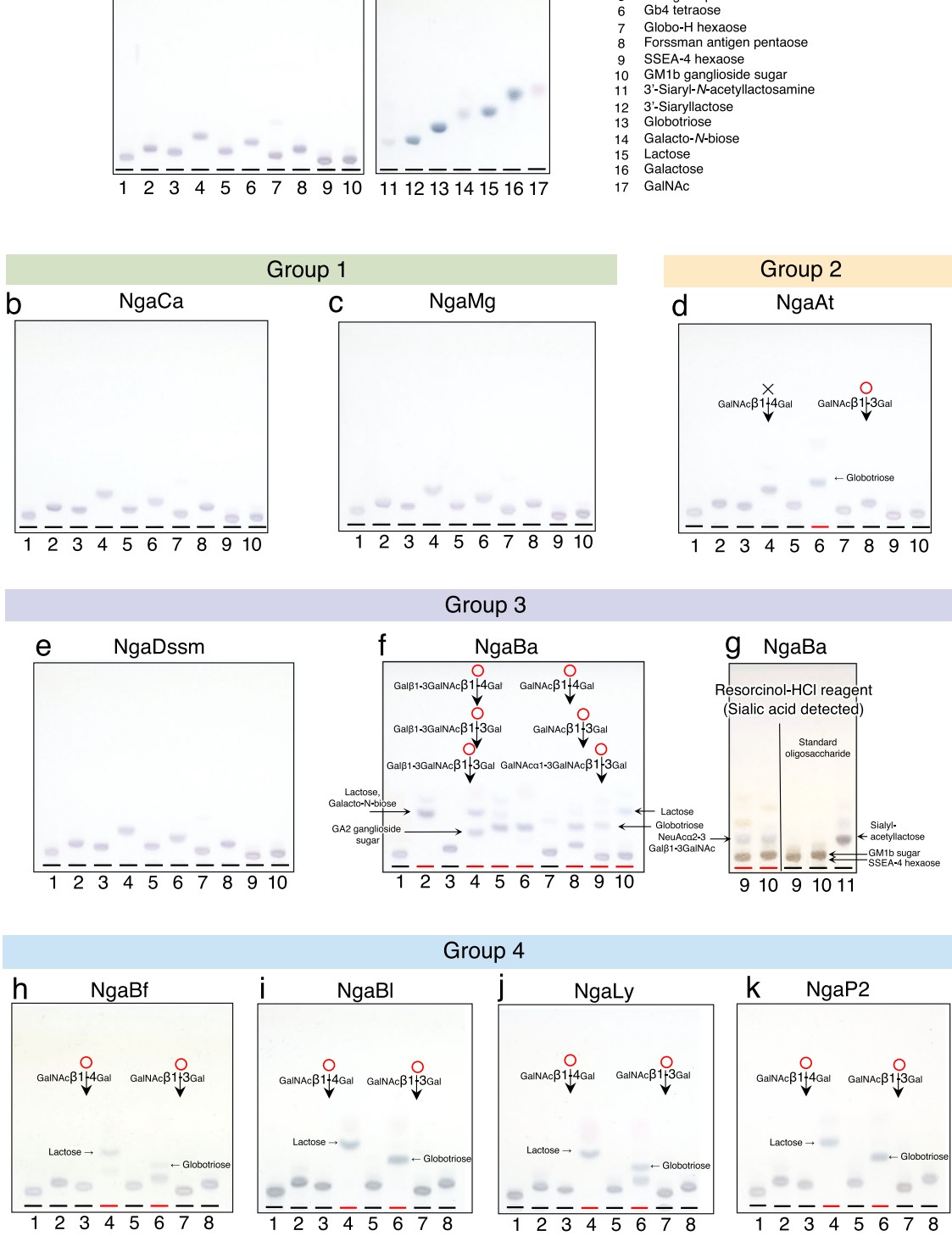

**Fig. 3 | Hydrolysis of oligosaccharides by recombinant β-NGAs. a** Thin layer chromatography analysis of the standard oligosaccharides and sugars. Lanes 1. GM1a ganglioside sugar; 2. GA1 ganglioside sugar; 3. GM2 ganglioside sugar; 4. GA2 ganglioside sugar; 5. Gb5 globopentaose; 6. Gb4 tetraose; 7. Globo-H hexaose; 8. Forssman antigen pentaose; 9. SSEA-4 hexaose; 10. GM1b ganglioside sugar; 11. 3′-Siaryl-*N*-acetyllactosamine; 12. 3′-Siaryllactose; 13. Globotriose; 14. Galacto-*N*-biose; 15. Lactose; 16. Galactose; 17. GalNAc. The structures and the Rf values of the standard oligosaccharides and sugar were listed in the Supplementary Table 1. **b–k** Thin layer chromatography demonstrates the hydrolysis of oligosaccharides by NgaCa (**b**), NgaMg (**c**), NgaAt (**d**), NgaDssm (**e**), NgaBa (**f**, **g**), NgaBf (**h**), NgaBl (**i**), NgaLy (**j**), NgaP2 (**k**). The oligosaccharides in each lane were arranged in the same order as in the standard TLC.

preference was reversed when *p*NP-substrates were used. Moreover, NgaBa displayed enzymatic activity against Galβ1-3GalNAc-β- and GalNAcα1-3GalNAc-β- and more effectively degraded Gb5 globopentaose than Forssman antigen pentaose (Fig. 3f and Supplementary Fig. 6b, d, lane 5 and 8). Furthermore, NgaBa did not act on Globo-H hexaose but could act on SSEA-4 hexaose and GM1b oligosaccharide (Fig. 3f and Supplementary Fig. 6b, d, lane 7, 9 and 10). Since the non-reducing terminal trisaccharides (NeuAcα2-3Galβ1-3GalNAcβ-) of SSEA-4 and GM1b were difficult to detect with diphenylamine-aniline-phosphate reagent, resorcinol-HCl reagent was used to detect the non-reducing terminal trisaccharides containing sialic acid owing to its specificity in detecting oligosaccharides containing sialic acid (Fig. 3g and Supplementary Fig. 6c, d, lane 9 and 10). Thus, NgaBa exhibited disaccharide-releasing activity and trisaccharide-releasing activity. None of the OR-β-NGAs acted on chondroitin sulfates A, B, and C, polymers containing sulfated GalNAc as a longer substrate polysaccharide (Supplementary Fig. 7). The NgaBf, NgaBl, NgaLy, and NgaP2 activities were similar to those of GH123 MR-β-NGAs, as they only degrade linear oligosaccharides with β-GalNAc at the non-reducing terminus (Fig. 3h–k). The results for NgaBf and NgaLy indicated that these enzymes preferred GalNAcβ1-4Gal to GalNAcβ1-3Gal. The natural substrates of NgaCa, NgaMg, and NgaDssm were not identified since only a few β-GalNAc-containing oligosaccharides are

commercially available, and GalNAc-containing oligosaccharides have diverse structures (Supplementary Fig. 1).

## X-ray crystal structures of the β-NGAs

Although the β-NGAs were divided into five major groups based on the phylogenetic analysis (including GH123 MR-β-NGAs), their predicted overall structure was similar. Based on substrate preferences, several of these enzymes considerably differed from GH123 MR-β-NGAs in terms of substrate specificity. To understand the relationship between structural variation and substrate specificity, crystallization screening was performed on all β-NGAs, and X-ray crystallographic analyses of apo- and/or ligand-bound forms were successfully performed on the following enzymes (NgaCa [Group 1], NgaAt [Group 2], NgaDssm [Group 3], NgaLy, and NgaP2 [Group 4]; nine forms of five enzymes in total) (Fig. 4 and Supplementary Tables 2–6). The overall structure of these enzymes is similar to that of CpNga123 and BvGH123, which consist of a β-sandwich domain at the N-terminus and a (β/α)8-barrel domain encompassing the catalytic region (Fig. 4a, Supplementary Figs. 8–12). The structure of NgaDssm (Group 3) bore a striking resemblance to that of NgaLy and NgaP2 (Group 4) (root-mean-square distance of Cα atoms [rmsd] = 1.5–2.3 Å), while the structures of NgaLy and NgaP2 are identical (rmsd = 1.0 Å) (Supplementary Table 7). The additional N-terminal domain consisting of

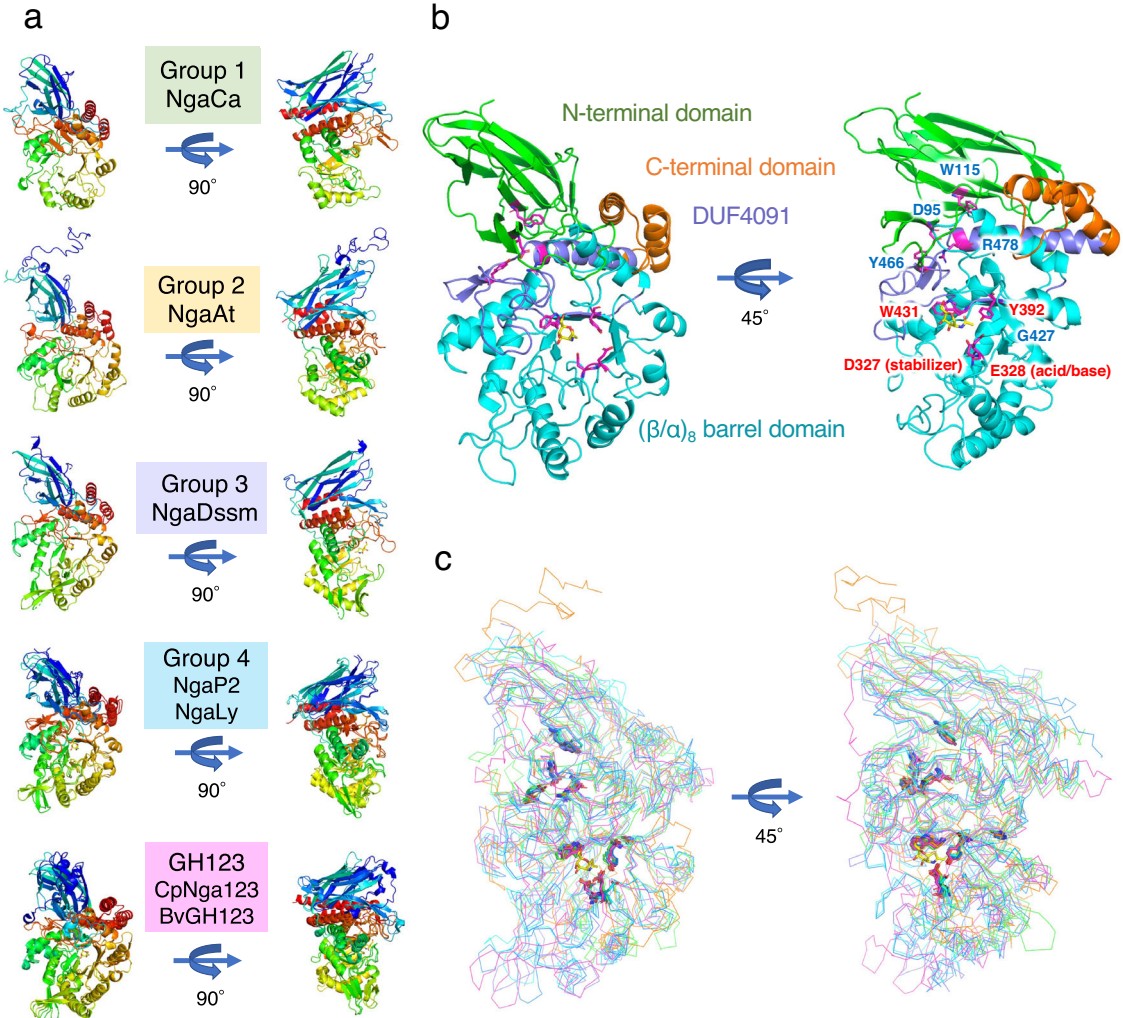

**Fig. 4 | Overall structures and superposition of β-NGAs. a** Structures of β-NGAs. The DE motif is shown in stick format. **b** The structures of NgaLy. The β-sandwich domain at the N-terminus, (β/α)8-barrel domain, DUF4091, and C-terminal domain are shown in green, cyan, purple, and orange, respectively. The conserved amino acids are shown as magenta sticks. Ligands are placed at the substrate-binding site and shown as yellow sticks. **c** Superposition of Group 1, Group 2, Group 3, Group 4 and GH123 structures. Conserved amino acids are plotted on the structures as sticks.

~80 amino acids, found only in Group 2 enzymes, was not modeled in the crystal structure of NgaAt owing to disorder (Supplementary Fig. 9, orange circle).

Among Groups 1–4 and GH123, the DE motif is located at the same position, suggesting that these enzymes adopt the substrate-assisted catalysis reported for GH123 MR-β-NGAs (Fig. 4b). The conserved DUF4091 domain in these enzyme groups is located at the innermost position of the enzyme between the N-terminal domain and the (β/α)₈-barrel domain (Fig. 4b, purple). This suggested that DUF4091 most likely plays a role in defining the location of N-terminal and (β/α)₈-barrel domains as a central pillar. Intriguingly, nine amino acid residues conserved among the groups (Fig. 1d, red background) are found at the same positions in the structure (Fig. 4c). Among the conserved amino acids, D95, W115, Y466, and R478 (amino acid numbers of NgaLy [Supplementary Table 8]) are located at the interface between the N-terminal domain and DUF4091 (Fig. 4b, right). D95 and W115, located at the N-terminal domain, fit into the surface pocket of DUF4091, and the side chain of D95 formed hydrogen bonds with Y466 and R478 located at DUF4091 (Fig. 5a). Therefore, D95, W115, Y466, and R478 are

potentially important for maintaining structural integrity as they link the N-terminal and (β/α)₈-barrel domains (Fig. 4b right, blue letter). G427 is positioned at the entry point for the eighth β-sheet into the (β/α)₈ barrel domain, with the space barely the size of the glycine residue (Fig. 5b). The DE motif (D327 and E328), Y392, and W431 are located around the substrate and are involved in substrate recognition (Fig. 4b right, red letter). Previous studies have reported that point mutations in the DE motif decrease catalytic activity[11,14]. Therefore, we examined the remaining seven conserved amino acids and constructed corresponding alanine mutants (Fig. 5c). The point mutants D95A, W115A, Y466A, and R478A were expressed in *E. coli*, but all proteins precipitated. For G427, a Val mutant was also constructed based on structural information suggesting that the conversion of Gly to Ala would be tolerated (Fig. 5b). In this case, a very small amount of solubilized enzyme was obtained for G427A but not for G427V (Fig. 5c, blue letter). The Y392A and W431A mutants were purified as soluble enzymes (Fig. 5c, red letter), although their catalytic activity was considerably reduced (Fig. 5d). These findings imply that amino acid substitutions severely impact the structural stability (D95, W115, Y466,

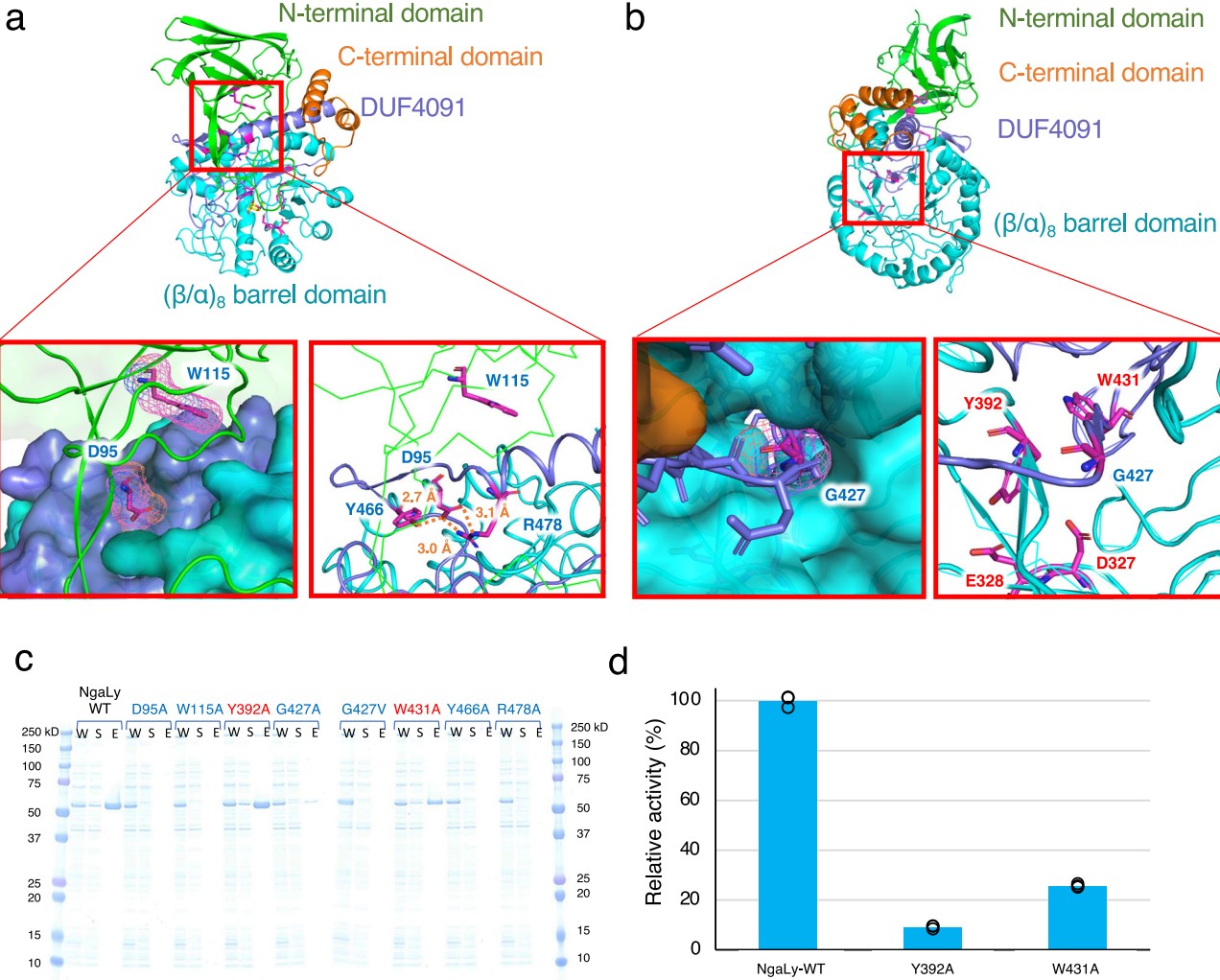

**Fig. 5 | Comparison of the overall structure and effect of the mutation on amino acids conserved throughout all β-NGA groups. a, b** Detailed views of interactions between conserved residues (**a** D95, W115, Y466, R478, **b** G427). The β-sandwich domain at the N-terminus, the (β/α)₈-barrel domain, DUF4091, and the C-terminal domain are depicted in green, cyan, purple, and orange, respectively. Magenta sticks represent the conserved amino acids. The ligands are indicated by yellow sticks. **c** SDS-PAGE analysis of point mutants (W: whole-cell lysate; S: supernatant

protein after sonication and centrifugation; E: eluted protein after affinity purification). The amino acids that likely contribute to structural stability and those involved in substrate recognition are demarcated with blue and red colors, respectively. **d** Relative activity of the NgaLy Y392A and W431A mutants compared with that of the wild-type (WT) enzyme. Values represent the mean of technical triplicate measurements.

R478, and G427) and substrate recognition (Y392 and W431) of the enzyme.

## Essential structural elements for OR- and MR-specificity

We compared the overall structure and substrate-binding sites among the groups to identify the structural elements governing OR- and MR-specificity of these enzymes (Fig. 6). NgaCa in Group 1 had the simplest structure among the enzymes analyzed (Fig. 6a, Supplementary Fig. 8). The active site of NgaCa possesses a cleft shape that allows oligosaccharides to pass through to the −2 subsite, enabling oligosaccharide binding and OR-type β-NGA activity. Furthermore, three characteristic tryptophan residues (W197, W218, and W249) are located in the subsites −2 and −3 (Supplementary Fig. 13a). Such a cleft structure and the presence of aromatic amino acids (such as

tryptophan or phenylalanine) in the substrate-binding site are characteristically observed in endo-type enzymes, such as chitinase or xyloglucanase which bind long oligosaccharides[35−37]. A docking model of NgaCa with oligosaccharide suggested that these tryptophan residues may play a role in a stacking-type interaction with sugars located in subsites −2 and −3. This structural feature indicated that Group 1 enzymes may act on the inner regions of longer sugar chains, in addition to disaccharides from the non-reducing end (e.g., Galβ1-3GalNAc-β-pNP).

The active site of NgaAt in Group 2 also possesses a cleft shape (Fig. 6b), and the second β-sheet of the (β/α)₈ barrel is longer than that found in the other enzymes, followed by an additional β-sheet (Fig. 6b and Supplementary Fig. 9, orange). Consequently, the space around the 3-OH group of β-GalNAc in the −2 subsite is narrower than that in

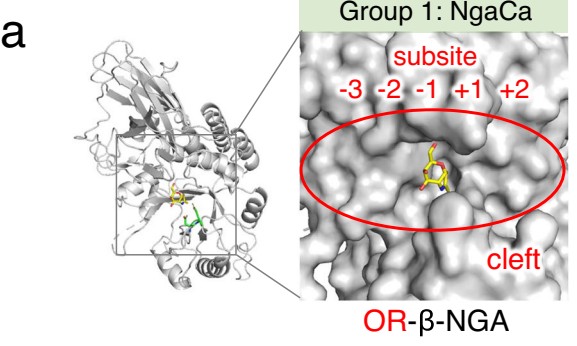

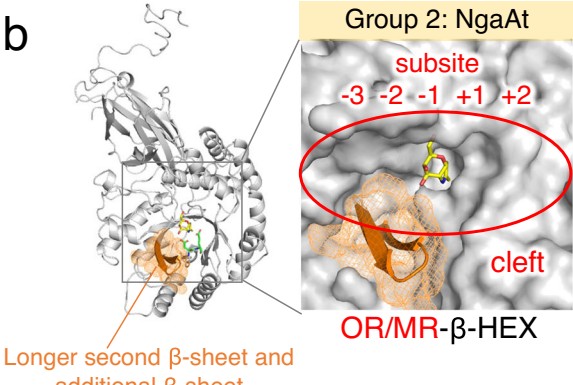

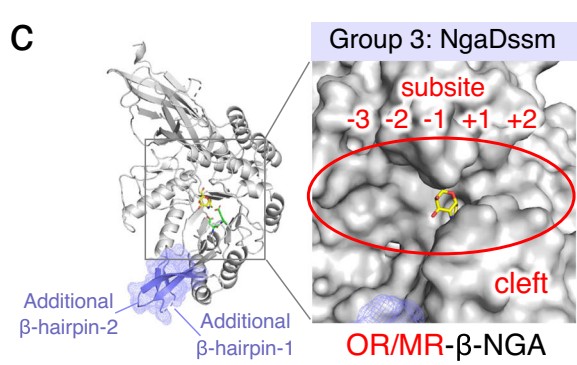

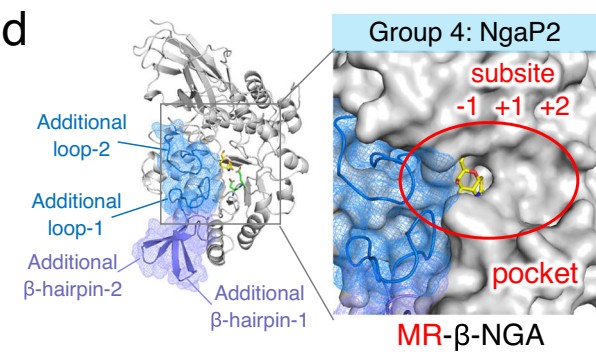

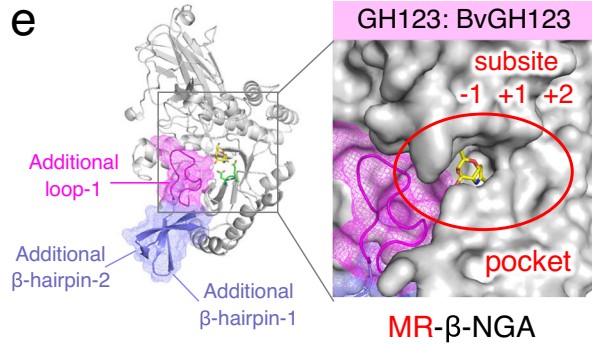

**Fig. 6 | Substrate-binding area of β-NGAs.** The overall structure of β-NGAs is shown as a cartoon (left) and the molecular surface of the substrate-binding site of β-NGA complexed with the ligand (right). The core structure (Group 1) is shown in gray (**a**), while additional regions characteristic of the other groups are shown in (**b**) orange (Group 2), (**c**) purple (Group 3), (**d**) blue (Group 4), and (**e**) magenta (GH123). The DE motif and ligand are indicated by green and yellow sticks, respectively.

Group 1. Therefore, NgaAt was presumed to act on Galβ1-4GalNAcβ- but not on Galβ1-3GalNAcβ-. Since Galβ1-4GalNAc-β-*p*NP was commercially unavailable, Galβ1-4GlcNAc-β-*p*NP was used as an alternative substrate to test this hypothesis, and we observed that Galβ1-4GlcNAc-β-*p*NP, but not Galβ1-3GalNAc-β-*p*NP, was degraded (Supplementary Fig. 4c). Thus, NgaAt demonstrates more diverse activity than expected and possesses OR/MR-β-HEX functionality.

The active site of the OR/MR-type NgaDssm from Group 3 has a cleft structure that enables passage to the −2 subsites, as in NgaCa, but it also possesses two additional β-hairpins from the second and third β-sheets of the (β/α)$_8$ barrel domain (Fig. 6c and Supplementary Fig. 10, purple). Moreover, NgaP2 (Group 4) has two loops above and below the cleft from the second and eighth β-sheets of the (β/α)$_8$ barrel domain (Fig. 6d and Supplementary Fig. 11, blue). The strict MR-type activity of Group 4 can be explained by these two loops that completely block the −2 subsite, forming a pocket-like architecture at the substrate-binding site and thus preventing substrate entry. NgNga belongs to Group 4 and has two additional loops that form a pocked-type substrate binding site, like NgaP2[16].

Similarly, BvGH123 in GH123 also has two additional β-hairpins and one extended loop from the second β-sheet of the (β/α)$_8$ barrel domain, which blocks the −2 subsite side of the cleft, yielding a pocket-like conformation (Fig. 6e and Supplementary Fig. 12, magenta).

These findings revealed that the substrate-binding sites with cleft structures exhibit OR-type activity, while those with pocket-like architectures show MR-type activity.

## Substrate recognition and catalytic mechanism of β-NGAs

We further examined detailed substrate recognition mechanisms using structural analysis of complexes with GalNAc-thiazoline, an analog of the oxazolinium intermediate and a potent inhibitor of enzymes utilizing substrate-assisted catalysis[38,39] (Fig. 7, Supplementary Fig. 14). With the exception of NgaCa, the structures of the NgaAt, NgaDssm, and NgaP2 complexes with GalNAc-thiazoline were successfully characterized (Supplementary Tables 3–5), and a subsequent docking model was engineered for NgaCa.

The docking model of NgaCa (Group 1) demonstrated that seven amino acid residues recognize GalNAc-thiazoline (Fig. 7a, green residues). In NgaAt (Group 2), apart from these seven residues, W329 establishes a hydrogen bond with GalNAc-thiazoline, and L555 contributes to substrate positioning (Fig. 7b, orange residues). Similarly, in NgaDssm (Group 3), the H479 residue forms a hydrogen bond, while residues W204 and L485 are involved in substrate positioning (Fig. 7c, purple residues). These results imply that Group 2 and Group 3 members possess a higher affinity for β-GalNAc than those of Group 1. Furthermore, complex binding modes were observed for Group 4 and GH123. For NgaP2 (Group 4), in addition to the seven residues discussed for NgaCa, three (D244, D297, and W298) and two (W196 and F489) residues are involved in hydrogen bond formation and substrate positioning, respectively (Fig. 7d, blue residues). Similarly, in BvGH123 (GH123), two (W253 and Q256) and three (W206, W306, and W482) residues are involved in these processes (Fig. 7e, magenta residues). Compared to the amino acid positions of the OR-type enzyme (green sticks), the MR-type enzymes have additional substrate-binding amino acid residues around β-GalNAc, indicating stronger recognition from the −2 subsite side. The increased number of residues could potentially reinforce substrate recognition in Group 4 and GH123 enzymes (Fig. 7f). Additional structural analyses (comparison of crystal structure with AlphaFold2 predicted structure of NgaCa, comparison of apo1-form and apo2-form of NgaCa, and comparison of GalNAc-thiazoline-bond form and GlcNAc-thiazoline-/apo-form of NgaAt, NgaDssm, and NgaP2) data are illustrated in Supplementary Fig. 13.

To complete the enzymatic analysis, we examined GalNAc- or GlcNAc-thiazoline as inhibitors of these enzymes. In the assays using GalNAc-thiazoline, Group 4 MR-β-NGAs showed inhibitory activity at

>1 nM concentrations (Fig. 7g, blue and light blue). By contrast, no inhibition was observed in Group 1 OR-β-NGA even at a concentration of 100 μM (Fig. 7g, green). This difference in the inhibitory activity of Group 1 enzymes can be explained as follows: the monosaccharide reaction intermediate-mimicking inhibitor (GalNAc-thiazoline) does not bind tightly to the substrate pocket of OR-type enzymes. Interestingly, the activity of Group 3 OR/MR-β-NGA was inhibited at concentrations 1000-fold higher than that for MR-β-NGA (Fig. 7g, purple). NgaAt was inhibited by 25 nM GalNAc-thiazoline and by 25 μM GlcNAc-thiazoline (a 1000-fold higher concentration) (Fig. 7g, orange and yellow). These results corroborate the disparity in recognition abilities of GalNAc-thiazoline identified from the aforementioned structures.

## Analysis of catalytic residue mutants

Based on structural analysis, Asp and Glu of the DE motifs were recognized as the stabilizer of the 2-acetamido group and acid-base catalytic residue, respectively. The GalNAc-thiazoline-bound structures and inhibition assays indicated that these enzymes perform substrate-assisted catalysis. Therefore, a point mutation analysis of the DE motifs was conducted (Fig. 7h, Supplementary Table 9). Mutants with Asp-to-Glu/Asn alterations exhibited little to no activity, even at 10-fold to 100-fold higher enzyme concentrations. Glu-to-Asp/Gln mutants demonstrated reduced enzymatic activities, and the trends were similar to those observed for other enzymes performing substrate-assisted catalysis[11,14].

## Elucidation of the catalytic mechanism using NMR

GH123 enzymes are MR-β-NGAs, while enzymes from Groups 1 and 3 are OR- and OR/MR-type β-NGAs, respectively. To further confirm the substrate-assisted catalysis mechanism of these enzymes, we investigated the reaction products by NMR using Galβ-1-3GalNAcβ-*p*NP as a substrate (Fig. 8, Supplementary Figs. 15–17, Supplementary Tables 10 and 11) and monitored the stereochemistry of glycosidic bond hydrolysis (between the GalNAc and *p*NP moieties) by $^1$H NMR. The anomeric hydrogen signal of Galβ-1-3GalNAcβ-*p*NP (between the GalNAc and *p*NP moieties) disappeared within 1 min for NgaDssm (Fig. 8a) and 10 min for NgaCa (Fig. 8b) and Galβ-1-3GalNAcβ appeared, while Galβ-1-3GalNAcα anomeric signals appeared after 10 and 30 min, respectively, due to mutarotation. These results indicate that NgaDssm and NgaCa are anomer-retaining enzymes, similar to GH123 enzymes.

## Discussion

We used deep-sea metagenomic sequences to discover a β-NGA, NgaDssm, which is the β-NGA to possess dual OR/MR-type-β-NGA activity and low sequence similarity to known MR-β-NGAs. Prior studies have characterized three GH123 MR-β-NGAs (NgaP, CpNga123, and BvGH123) from land soil[11] and human gut bacteria[14,15]; however, no genes from other ecological niches have been reported. The deep-sea environment (below 200 m depth) is characterized by total darkness, low temperatures, and high pressure, with occasional high temperatures owing to geological formations, such as hydrothermal vents. The deep-sea environments are completely distinct from the terrestrial environment and remain unexplored owing to limited accessibility to samples. Therefore, the deep-sea microbiome is an attractive potential bioresource for screening undiscovered enzymes.

We further discovered OR-, OR/MR-, and MR-type β-NGAs, as well as OR/MR-β-HEXs, acting on β-GalNAc by analyzing deep-sea metagenomic sequences and public protein databases, and our comparative biochemical and structural analyses provide insights into the molecular evolution of β-GalNAc-targeting enzymes (Fig. 9). Despite their phylogenetic distance and different substrate specificities, these enzymes and GH123 β-NGAs are likely homologous proteins based on the presence of conserved residues in their sequences (Fig. 1d), which is also positionally retained in their structures (Fig. 4c), and the observation that mutations in conserved residues were not tolerated

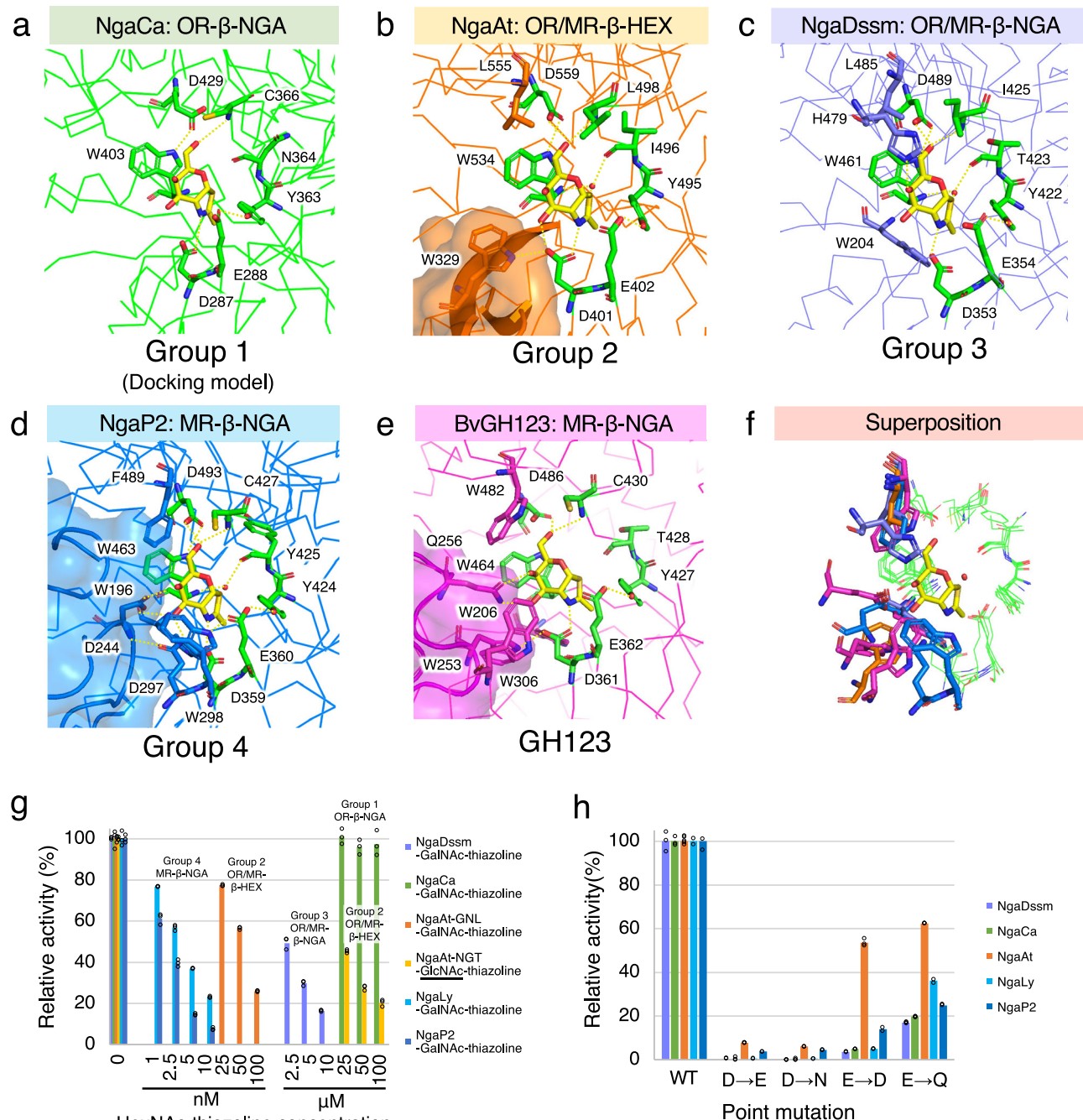

**Fig. 7 | Substrate recognition and catalytic mechanism. a−e** Active site structure of each enzyme. **a,** Docking model of NgaCa with GalNAc-thiazoline. **b−e** Crystal structure of each enzyme complexed with GalNAc-thiazoline. Hydrogen bonds are indicated by dotted lines. **f** Superimposition of active sites in Groups 1, Group 2, Group 3, Group 4 and GH123. **g** Inhibition of β-NGA activity by GalNAc-thiazoline and GlcNAc-thiazoline. **h** Point mutation analysis of the DE motif. The amounts of enzyme used in the point mutation assay were listed in Supplementary Table 9.

and resulted in destabilization of protein structure and elimination of substrate recognition (Fig. 5c, d). Structural comparisons between members in Groups 1–4 and GH123 enzymes further supported that the five families have diversified their substrate specificity (OR-β-NGA for Group 1, OR/MR-β-HEX for Group 2, OR/MR-β-NGA for Group 3, MR-β-NGA for Group 4, and MR-β-NGA for GH123) through the accumulation of point mutations and insertional sequences (Figs. 6 and 7). These data suggested a monophyletic evolutionary history of β-NGAs from the prototype enzymes in Group 1 β-NGAs (Fig. 9).

For the family division of the four groups of enzymes identified in this study, the CAZy team's bioinformatics perspective on the sequence was concluded as follows: there are subfamilies (Groups 3 and 4) closer to the main (sub)family, GH123, and distant groups (Groups 1 and 2). The distant subgroups (Groups 1 and 2) exhibit highly distinct sequence profiles that could potentially be categorized as distinct families. However, given their size and functional conservation, they were designated as a distant subfamily. Following these decisions, GH123 was reconstituted as a large GH family with several subfamilies. Among the enzymes identified herein, only NgaAt (Group 2) is a β-HEX that acts on β-GalNAc and β-GlcNAc [NgaAt (locus_tag = AT1G45150) is currently annotated as an α-1,6-mannosyl-glycoprotein 2-beta-*N*-acetylglucosaminyltransferase (EC 2.4.1.143); this annotation needs correcting based on our findings]. We speculate that a prototype enzyme of the series of β-NGAs, including Group 1–4 and GH123, were

## a  NgaDssm

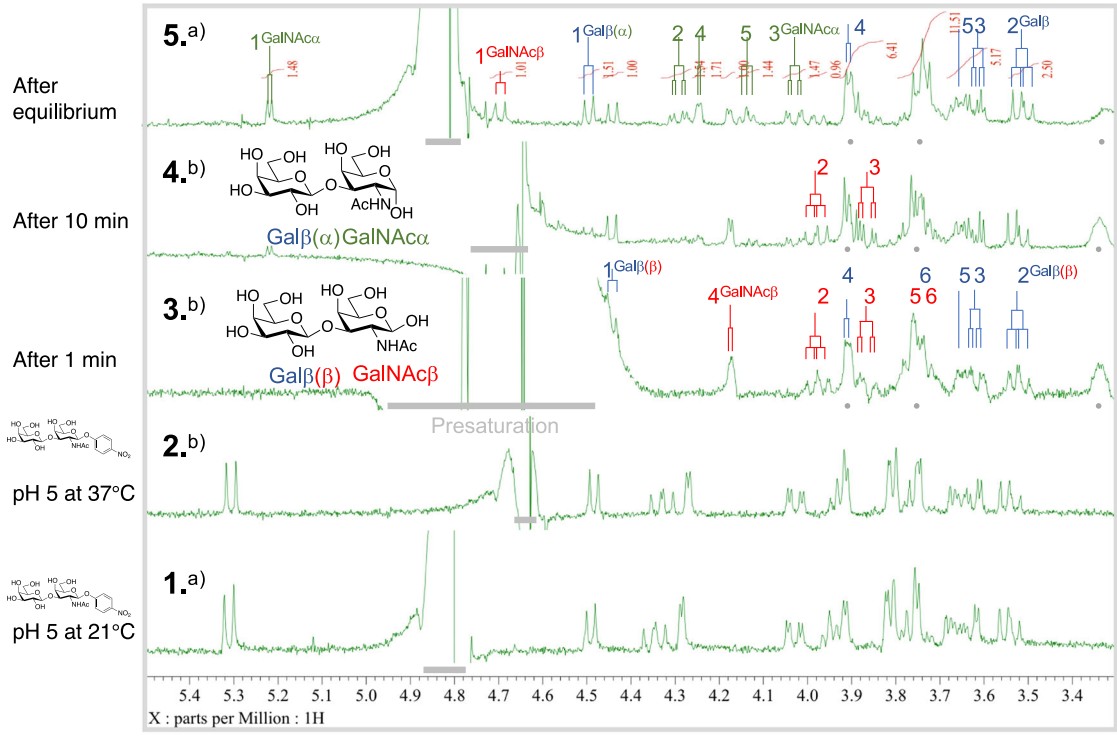

## b  NgaCa

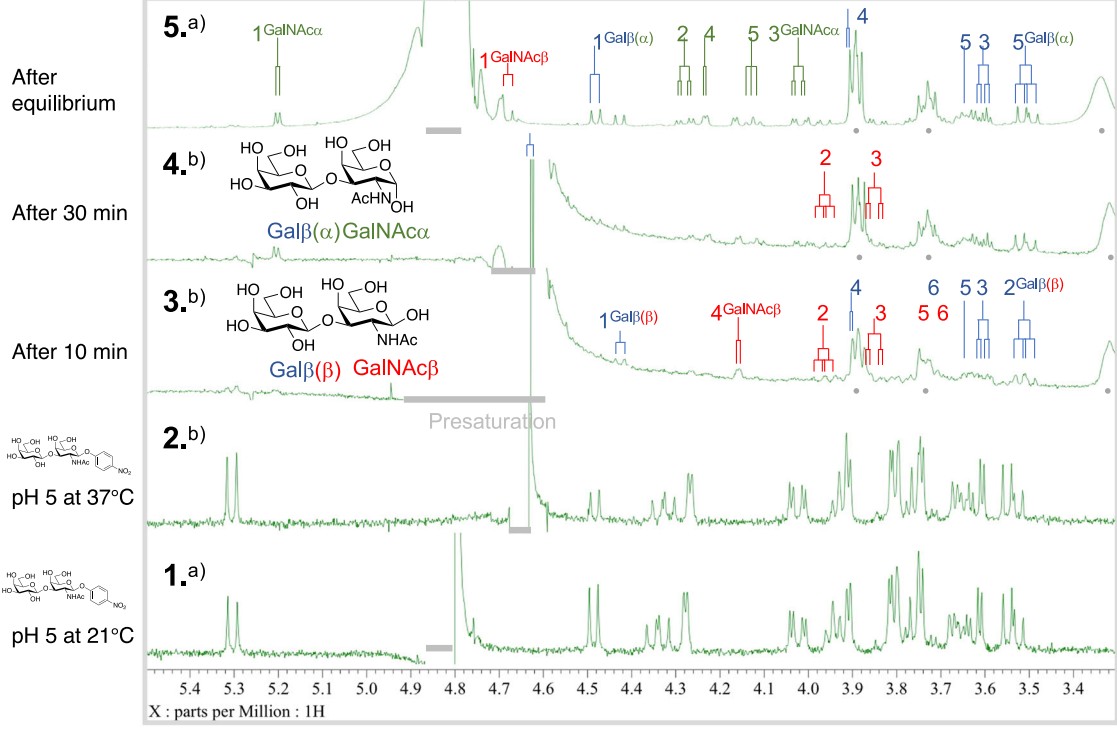

**Fig. 8 | NgaDssm and NgaCa are anomer-retaining enzymes. a** [1]H NMR spectrum monitoring the activity of NgaDssm toward Galβ1-3GalNAc-β-*p*NP in $D_2O/H_2O$ (32:1, 100 mM citrate buffer, pH(D) 5.0) at 37 °C. 1. Substrate at 21 °C; 2. Substrate at 37 °C at pH(D) 5.0 before the addition of the enzyme solution; 3. Reaction after 1 min; 4: Reaction after 10 min; 5: Reaction after attaining equilibrium. The enzyme is dissolved in 20 mM HEPES-Na (pH 7.5), 150 mM NaCl, and 1 mM DTT in $H_2O$ and premixed with $D_2O$ ($D_2O/H_2O$ = 6:1) before treatment with the substrate. **b** [1]H NMR spectrum monitoring the activity of NgaCa toward Galβ1-3GalNAc-β-*p*NP in $D_2O/$ $H_2O$ (11:1, 100 mM citrate buffer, pH(D) 5.0) at 37 °C. 1. Substrate at 21 °C; 2. substrate at 37 °C $D_2O/H_2O$ (11:1, 100 mM citrate buffer, pH(D) 5.0); 3. reaction after 10 min; 4. reaction after 30 min; 5. reaction after attaining equilibrium. The enzyme is dissolved in 20 mM HEPES-Na (pH 7.5), 150 mM NaCl, and 1 mM DTT in $H_2O$ (32.5 mg/mL) and premixed with $D_2O$ ($D_2O/H_2O$ = 2:1) before treatment with the substrate. **a** Measured at 21 °C with presaturation at 4.80 ppm. **b** Measured at 37 °C with presaturation at 4.63 ppm. The gray dots symbolize the peaks of the reagents in the solution (DTT and HEPES). The gray bars indicate the areas affected by presaturation to reduce the HOD peak.

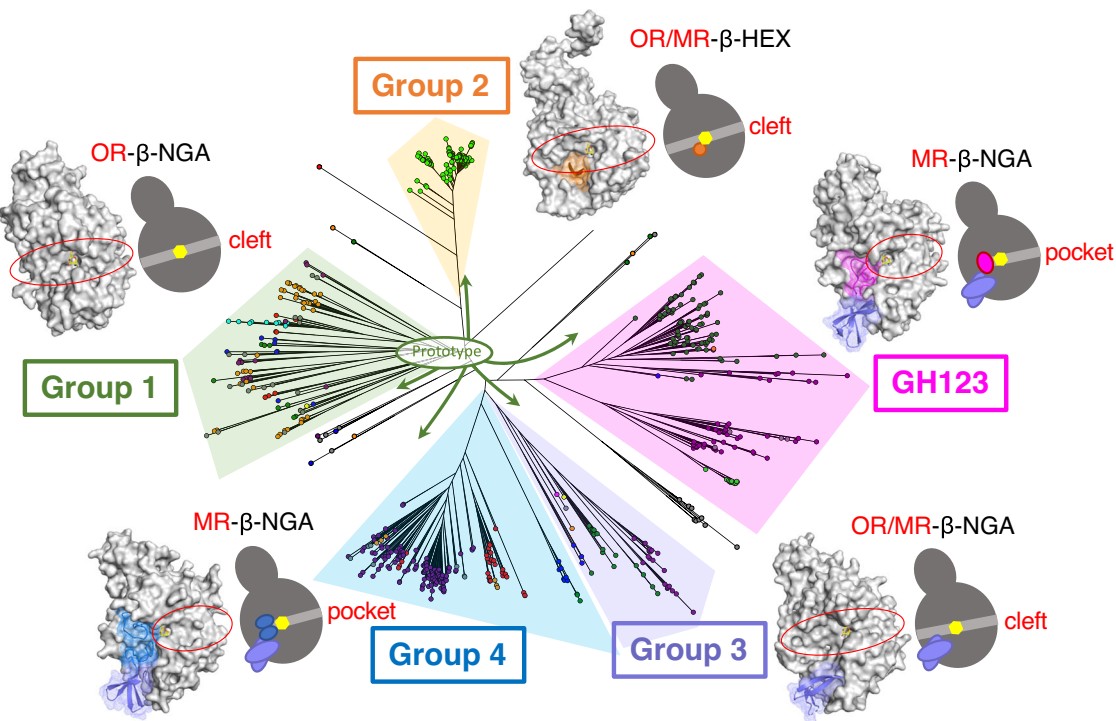

**Fig. 9 | Phylogenetic tree and structural features of β-NGAs.** The overall structure of β-NGAs is illustrated as a surface model with an accompanying schematic representation. The ligand is indicated by a yellow stick. The Group 1 structure is depicted in gray as the basic structure. Additional regions characteristic of other groups are highlighted in orange (Group 2), purple (Group 3), blue (Group 4), and magenta (GH123).

a β-NGA specific for β-GalNAc, and the β-HEX enzymes of Group 2 are proposed to be divergent from the structurally simpler β-NGA. By contrast, GH20, mainly comprising β-HEX enzymes, belongs to clan GH-K, together with GH18 and GH85. All GH-K enzymes act on β-GlcNAc bonds (e.g., exo- and endo-β-*N*-acetylglucosaminidase, chitinase, lacto-*N*-biosidase, and hyaluronidase). Thus, GH-K enzymes seem to originate from a structurally simple β-*N*-acetylglucosaminidase, and GH20 β-HEX appears to belong to a family that diverged from the prototypical β-*N*-acetylglucosaminidase. Therefore, since β-HEX acts on β-GalNAc and β-GlcNAc, we suggest that β-HEX has two enzyme lineages with different evolutionary pathways.

In a previous study on β-NGAs from GH123, the function of DUF4091 has never been reported[14,15]. However, our domain-based exploration of β-NGA genes, based on GH123 members in the deep-sea metagenomic data, led to the identification of another lineage of β-NGAs containing DUF4091. Mutation experiments also highlighted that DUF4091 is indispensable for the stability of β-NGAs. The functional annotation of CDSs is typically conducted based on the similarity of the entire sequence to known protein sequences. A domain-based search (typically based on hidden Markov models) is an alternative method for identifying homologous sequence regions, even if the overall similarity is low. Examining CAZy families with consideration of co-occurring functionally unknown domains will likely lead to the subsequent discovery of additional enzymes.

To investigate the possible biological functions of the β-NGAs, we examined neighboring genes and protein-protein interactions using the STRING database[40] (Supplementary Fig. 18, Supplementary Data 4 and 5). The enzymes identified herein are potentially involved in the regulation of lipoprotein (NgaCa), the capsular membrane (NgaCp), and O-antigen (NgaCs), as well as in the degradation of glycans in the periplasm (NgaBf). These results suggest that OR-β-NGA may be involved in the degradation of long-chain glycoconjugates with repeating units of exopolysaccharides (Supplementary Fig. 1). Although the actual biological substrates of OR-β-NGAs such as NgaCa

were not identified in this study, owing to the diversity of these glycan structures and the variation among species, considering the possibility that the actual natural substrates of OR-β-NGA are such long-chain repeating oligosaccharides, the substrate binding site of OR-β-NGA, with a cleft structure and characteristic aromatic amino acids, would serve as a clue to uncovering the biological role of this enzyme. This observation is consistent with the hypothesis that it binds long-chain oligosaccharides. Although the OR-β-NGAs reported here did not directly hydrolyze chondroitin sulfate (Supplementary Fig. 7), these β-NGAs, together with β-glucuronidase and sulfatase, might play a role in the degradation of chondroitin sulfate oligosaccharides, which are typically degraded by chondroitin sulfate lyase[41]. The natural substrate of NaNga from *N. aurantiaca* also has not been identified, although the possibility of the exopolysaccharide arabinogalactan has been suggested[16]. Therefore, β-NGAs may exhibit more diverse functions, including roles in various GalNAc-mediated biological processes in addition to the degradation and utilization of glycans shown for GH123 β-NGA.

Since the majority of prokaryotes in nature remain uncultured, it is crucial to explore the functional and genetic diversity of glycosidases in uncultured microorganisms through metagenomic analysis in order to gain a comprehensive understanding of glycan-mediated phenomena. The glycosidases hold potential for various industrial applications, including the utilization of these enzymes as biocatalysts for the production of functional oligosaccharides, glycan structure analysis, and disease diagnosis[33]. Therefore, the discovery of various glycosidases is crucial for both the basic and applied sciences. The comprehensive exploration and characterization of the diverse β-NGAs presented significantly enhance our understanding of their biological functions and their evolutionary history. These findings present an approach to enunciating the evolutionary history of not only β-NGA but also many glycan-related enzymes. Further elucidation of the correlation between structures and diverse functions determined during evolution would also have significant implications for

the structural basis of enzyme design for the engineering of specific enzymes.

## Methods

### Sediment sampling and metagenomic sequencing

Abyssal sediment core was collected using a gravity corer on a remotely operated vehicle *ABISMO*[42] at station IOB (located 29.2746° N, 143.7673° E, at a depth of 5747 m below sea level) during a cruise of the ship R/V *Kairei* KR11-11 (December 2011) owned by the Japan Agency for Marine-Earth Science and Technology (JAMSTEC). The acquired sediment core was immediately subsampled onboard and stored at −80 °C for molecular biology analyses[43].

Approximately 5 mL of frozen subsampled sediments from four sections (extracted from the depth ranges 0–8, 13–23, 53–63, and 113–123 cm below the seafloor) were used for metagenomic analysis. The sections were selected based on the geochemical profile and microbial community composition of the sediment core[43]. Environmental DNA was extracted using the DNeasy PowerMax Soil Kit (QIAGEN, Hilden, Germany) according to the manufacturer's protocol, with the following minor modification to increase DNA yield: cells were agitated twice for 10 min each after incubation at 65 °C. Sequence libraries were prepared using the KAPA HyperPrep Kit (KAPA Biosystems) or the Ovation SP+ Ultralow Library System (NuGEN Technologies, San Carlos, CA, USA)[44]. Library pools were mixed with Illumina PhiX control libraries and sequenced using the Illumina MiSeq or HiSeq platforms (Illumina, San Diego, CA, USA) at JAMSTEC or Macrogen (Seoul, South Korea).

### Bioinformatics

For raw metagenomic sequence data, both ends of the reads containing low-quality bases (Phread quality score <20) and adapter sequences were trimmed using TrimGalore (https://github.com/FelixKrueger/TrimGalore) with default settings. Sequencing reads derived from the PhiX genome was removed using Bowtie2[45]. Low complexity sequences or those shorter than 100 bp were discarded using PRINSEQ++[46]. The remaining high-quality paired-end reads of each sample were individually assembled de novo using metaSPAdes[47]. Full-length coding sequences (CDSs) in the contigs were predicted using Prodigal[48] in the anonymous mode ('-p meta' setting). Protein domain annotations of the CDSs were achieved through HMMER[49] against Pfam (version 35.0)[50] and dbCAN2 (version v10)[51] with a cutoff domain *e*-value of ≤ 1E−3. All CDSs assigned to GH123 and measuring >400 bp in length were retrieved as β-NGA candidates and used for further analysis.

In addition to metagenomic CDSs, those with >400 bp and architecture similar to known β-NGA genes (containing the DUF4091 [PF13320] domain in the C-terminal region) were retrieved from Pfam (version 35.0). A phylogenetic tree was constructed using MAFFT[52] with default settings and FastTree2[53] with JC + CAT models. ClustalOmega[54] and the ColabFold software[55] were employed for sequence alignment and protein structure prediction, respectively.

### Construction of β-NGA expression vectors

The selected β-NGA candidates were artificially synthesized by codon optimization for recombinant expression in *E. coli* using Strings DNA Fragments Synthesis Service (Thermo Fisher Scientific, Waltham, MA, USA) (Supplementary Data 3). The signal peptides in the amino acid sequences (Supplementary Data 1 and 3, highlighted with bold and underlined font) were predicted using SignalP 5.0[56] and were removed from the N-terminal side of NgaDssm, NgaNp, NgaCs, NgaSc, and NgaCp. The gene encoding NgaP2 was directly cloned from the genomic DNA of *Paenibacillus* sp. TS12 via PCR amplification. These synthetic genes were designed with additional sequences at both terminals to facilitate amplification using a common primer set

(Supplementary Data 6). The genes were cloned into the pET-47b(+) expression vector (Merck KGaA, Darmstadt, Germany) using the In-Fusion HD Cloning Kit (Takara Bio, Shiga, Japan). Mutagenesis was performed using the In-Fusion HD Cloning Kit. Primer sequences are listed in Supplementary Data 6.

### Expression and purification of recombinant β-NGA

The expression vector and recombinant mutant plasmids were used to transform *E. coli* BL21 Star (DE3) cells. The cells were cultured in 50 mL of medium A (LB medium containing 50 µg/mL of kanamycin), incubated at 37 °C for 16 h with shaking, inoculated into 1–2 L of medium A, and incubated at 37 °C for another 2–3 h with shaking. Protein expression was induced by the addition of isopropyl β-D-1-thiogalactopyranoside (IPTG) to the culture at a final concentration of 0.1 mM. After additional culturing at 16 °C for 16 h, cells were harvested by centrifugation (10,000×*g* for 10 min) and suspended in 50 mL of buffer A (20 mM HEPES-Na [pH 7.5], 150 mM NaCl, 5% [v/v] glycerol, 1 mM dithiothreitol (DTT), and 50 mM imidazole). Following sonication, cell debris was removed by centrifugation (13,000×*g* for 20 min at 4 °C) and passed through a 0.45-µm pore-sized GD/X syringe filter (Cytiva, Marlborough, MA, USA). The supernatant was subjected to chromatography using the ÄKTA Prime Chromatography System (Cytiva). The sample was loaded onto a 5 mL HisTrap HP column (Cytiva) at a flow rate of 2 mL/min. The column was then washed with buffer A. The His-tagged protein was eluted with Buffer B (20 mM HEPES-Na [pH 7.5], 150 mM NaCl, 5% [v/v] glycerol, 1 mM DTT, and 300 mM imidazole). The eluted fractions were pooled and dialyzed against Buffer C (20 mM HEPES-Na [pH 7.5], 150 mM NaCl, and 1 mM DTT). To cleave the His-Tag, the HRV3C protease was dialyzed at 4 °C for 16 h. The enzyme was further purified by ion-exchange [5-mL HiTrapQ column (Cytiva)] and size exclusion chromatography [HiLoad 16/600 Superdex 200 pg column (Cytiva)]. The presence of the desired protein was confirmed by SDS-PAGE. The molecular weights of all β-NGAs estimated by SEC were consistent with the calculated molecular weights of their monomers.

### Enzyme assays

The activity of β-NGA candidates was determined using Assays I–III. In Assay I [GalNAc-β-*p*NP (Sigma-Aldrich) as a substrate], the reaction mixture comprised 50 nmol of GalNAc-β-*p*NP and an appropriate amount of the enzyme in 100 µL of a 100 mM optimal pH buffer solution listed in Table 1. The reaction was carried out at the optimal temperature listed in Table 1 for 30-min. The reaction was arrested by adding 100 µL of 1 M sodium carbonate, and the corresponding absorbance was measured at 405 nm. The time course indicates that the 30-min assays were measured at the initial rate (Supplementary Fig. 19). One unit of the enzyme was defined as the amount that catalyzed the release of 1 µmol of *p*-nitrophenol per min from GalNAc-β-*p*NP under experimental conditions. Values represent the mean of technical triplicate measurements. Controls were measured without enzyme as the blank, and relative activity was calculated as 100% of the activity of GalNAc-β-*p*NP (or Galβ1-3GalNAc-β-*p*NP for NgaCa). The optimal pH was determined using the reaction mixture comprising 50 nmol of GalNAc-β-*p*NP (or Galβ1-3GalNAc-β-*p*NP for NgaCa) and an appropriate amount of the enzyme in 100 µL of a 100 mM GTA buffer [50 mM 3,3-dimethyl glutaric acid, 50 mM tris(hydroxymethyl)aminomethane, and 50 mM 2-amino-2-methyl-1,3-propanediol]. Following incubation at 37 °C for 30-min, the reaction was arrested by adding 100 µL of 1 M sodium carbonate, and the corresponding absorbance was measured at 405 nm. The effect of metal ion was determined using the reaction mixture comprised 50 nmol of GalNAc-β-*p*NP (or Galβ1-3GalNAc-β-*p*NP for NgaCa) and an appropriate amount of the enzyme in 100 µL of a 100 mM optimal buffer solution listed in Table 1.

Following incubation at 37 °C for 30-min, the reaction was arrested by adding 100 μL of 1 M sodium carbonate, and the corresponding absorbance was measured at 405 nm. The optimal temperature was determined using the reaction mixture containing 50 nmol of GalNAc-β-*p*NP (or Galβ1-3GalNAc-β-*p*NP for NgaCa) at various temperatures (in increments of 5 °C) and an appropriate amount of the enzyme in 100 μL of a 100 mM optimal buffer solution listed in Table 1. Following incubation for 30-min, the reaction was arrested by adding 100 μL of 1 M sodium carbonate and measured as described above. The optimal temperature was determined using the reaction mixture containing 50 nmol of GalNAc-β-*p*NP (or Galβ1-3GalNAc-β-*p*NP for NgaCa) at various temperatures (in increments of 5 °C) and an appropriate amount of the enzyme in 100 μL of a 100 mM optimal buffer solution listed in Table 1. Following incubation for 30-min, the reaction was arrested by adding 100 μL of 1 M sodium carbonate and measured as described above. To avoid thermal degradation of *p*NP-substrates, the optimum temperatures of the enzymes were maintained between 0 and 75 °C. The substrate specificity of the enzymes was examined using the following *p*NP-glycosides (50 nmol) (Sigma-Aldrich) or 4MU-glycosides (10 nmol) (Sigma-Aldrich): GalNAc-β- or α-*p*NP, Galβ1-3GalNAc-β- or α-*p*NP, GlcNAc-β- or α-*p*NP, galactose-β- or α-*p*NP, glucose-β- or α-*p*NP, arabinose-β- or α-*p*NP, mannose-β- or α-*p*NP, fucose-β- or α-*p*NP, xylose-β- or α-*p*NP, sulfate-*p*NP, GalNAc-β-4MU, GalNAc4S-β-4MU and GalNAc6S-β-4MU. Assay II [GalNAc-β-4MU (Sigma-Aldrich) as a substrate], the reaction mixture was formulated using 10 nmol of GalNAc-β-4MU and an appropriate amount of enzyme in 100 μL of a 100 mM buffer solution listed in Table 1. Following incubation for 30 min, the fluorescence intensity was measured using a Synergy 2 multimode microplate reader (BioTek) at excitation and emission wavelengths of 360 and 460 nm, respectively. Values represent the mean of technical triplicate measurements. In Assay III (oligosaccharides as substrates), reaction mixtures containing 5 nmol of oligosaccharides and an appropriate amount of enzyme in 20 μL of 100 mM buffer [sodium acetate buffer (pH 5.0–5.5), Bis-Tris buffer (pH 6.0–7.5), or Tris-HCl buffer (pH 8.0)] corresponding to the optimal pH of each enzyme were incubated at optimal temperature for 16 h. The samples were boiled for 5 min to stop the reaction. The samples were dried, and the residues were dissolved in 10 μL of a methanol:water (1:1, v/v) solution and applied to a precoated Silica Gel 60 TLC glass plate (Merck), which was developed using a 1-butanol:acetic acid:water (2:1:1, v/v/v) solution. Oligosaccharides and GalNAc were visualized using a diphenylamine-aniline-phosphate reagent [a mixture of 0.4 g diphenylamine, 0.4 mL aniline, 3 mL 85% phosphoric acid and 20 mL acetone]. Sialic acid-containing oligosaccharides were visualized using resorcinol-HCl reagent [0.2 g of resorcinol was dissolved in 10 mL water and then added to 80 mL of HCl and 0.25 mL of 0.1 M copper sulfate. Finally, the reagent was made up to 100 mL with water]. The reaction products were identified by comparing them with the Rf value of the standard oligosaccharides and sugar. The oligosaccharides [GM1a ganglioside sugar (ELICITYL, Crolles, France), GA1 ganglioside sugar (ELICITYL), GM2 ganglioside sugar (ELICITYL), GA2 ganglioside sugar (ELICITYL), Gb5 globopentaose (ELICITYL), Gb4 tetraose (Biosynth, UK), Globo-H hexaose (ELICITYL), Forssman antigen pentaose (ELICITYL), SSEA-4 hexaose (ELICITYL), GM1b ganglioside sugar (ELICITYL), 3'-siaryl-*N*-acetyllactosamine (Biosynth), 3'-sialyllactose (TCI, Tokyo, Japan), globotriose (TCI), galacto-*N*-biose (Biosynth)], chondroitin sulfate A (Biosynth), chondroitin sulfate B (FUJIFILM Wako Pure Chemical Corporation, Osaka, Japan), chondroitin sulfate C (FUJIFILM), ΔDi-4S (Iwai Chemicals Company, Ltd. Tokyo, Japan), and chondroitinase ABC from *Proteus vulgaris* (Sigma-Aldrich) were purchased.

## Protein thermal shift assay

The protein thermal shift assay was conducted using the StepOnePlus Real-Time PCR System (Applied Biosystems) with Applied Biosystems Protein Thermal Shift Dye. The $T_m$ of the proteins was calculated using the Protein Thermal Shift software v1.4 (Applied Biosystems).

## Crystallization, data collection, and structure determination

Purified β-NGA was concentrated at 10–20 mg/mL. This sample solution (0.5 μL) was mixed with an equivalent volume of the reservoir solution. Crystallization was performed by the sitting drop vapor diffusion technique at 20 °C. Crystals were initially formed in the crystallization screening trial and were reproduced by seeding the crystals into solutions prepared in an identical manner (Supplementary Data 7). For crystallization of the β-NGA-(GalNAc-thiazoline) complex, GalNAc-thiazoline was incorporated into the β-NGA protein solution to obtain a final concentration of 5 mM. GalNAc-thiazoline was prepared as described in detail by Knapp and coworkers[38]. A crystallization solution containing 20% (v/v) glycerol was used as a cryoprotectant for X-ray diffraction data collection. The X-ray diffraction experiments were performed at the BL32XU beamline of SPring-8. All diffraction data were collected using the automated data collection system ZOO[57]. The obtained data were processed with XDS[58] using the automated data processing pipeline KAMO[59]. For the X-ray diffraction data of NgaDssm complexed with GalNAc-thiazoline, automated structural analysis was performed using NABE[60], and the structure was solved by molecular replacement using the AlphaFold2 model[55]. PHENIX[61], COOT[62], and REFMAC[63] were employed for structure refinement. Molecular images were displayed using PyMol (Schrödinger LLC, Palo Alto, CA, USA). The secondary structural elements in Supplementary Figs. 8–12 were determined using the ESPript software[64].

**NMR experiments.** $^1$H NMR and $^{13}$C NMR spectra were recorded on a JEOL ECX400 spectrometer and processed by Delta ver. 6.0 software. $^1$H NMR spectra were referenced to HOD at 4.80 ppm (21 °C) and at 4.63 ppm (37 °C). $^{13}$C NMR spectra were referenced to the native scale unless otherwise mentioned. Assignments were made by standard pfg COSY, pfg HMQC or pfg HMBC, and so on. To NMR sample tube (PS-003-7, Shigemi Co., Ltd.), 140 μL of 5.0 mM solution of Galβ1-3GalNAc-β-*p*NP and 70 μL of 1 M sodium citrate pH 5.0 in D$_2$O was added with 340 μL of D$_2$O. To the mixture, a mixture of NgaDssm enzyme solution (21 μL in 20 mM Hepes-Na pH 7.5, 150 mM NaCl, and 1 mM DTT in H$_2$O, which is the calculated amount for the 100% conversion after 1 min) in H$_2$O and 129 μL of D$_2$O was added. Then the mixture was set into an NMR machine, which was preoptimized for shim using the same solvent mixture with substrate and heated to 37 °C. $^1$H NMR spectra were obtained for monitoring by keeping the temperature at 37 °C in the NMR machine (Fig. 8a). To another NMR sample tube, 120 μL of 5.0 mM solution of Galβ1-3GalNAc-β-*p*NP and 60 μL of 1 M sodium citrate pH 5.0 in D$_2$O was added with 270 μL of D$_2$O. To the mixture, a mixture of NgaCa enzyme solution (50 μL in 20 mM HEPES-Na pH 7.5, 150 mM NaCl and 1 mM DTT in H$_2$O (32.5 mg/mL), which is the calculated amount for the 15% conversion after 1 min) in H$_2$O and 100 μL of D$_2$O was added. Then the mixture was set into the NMR machine, heated to 37 °C, and monitored (Fig. 8b).

## Reporting summary

Further information on research design is available in the Nature Portfolio Reporting Summary linked to this article.

# Data availability

The metagenomic sequencing data were deposited in the DDBJ Sequence Read Archive under BioProject ID PRJDB15058. All coordinates are deposited in the PDB under accession numbers 8K2F, 8K2G, 8K2H, 8K2I, 8K2J, 8K2K, 8K2L, 8K2M, and 8K2N. STRING database is available here: [https://version-11-5.string-db.org]. Input protein names are listed in Supplementary Data 5. All the requisite data for evaluating the conclusions are present in the article and Methods. Source data are provided with this paper.

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

## Acknowledgements

We express our sincere appreciation to the captain, crew, and all onboard scientists and technicians of the KR11-11 cruise. We are grateful to the ROV *ABISMO* development and operation teams. Computations were partially performed on the NIG supercomputer at the ROIS National Institute of Genetics and the Data Analysis System and the Earth Simulator at JAMSTEC. This research was funded by the Research Support Project for Life Science and Drug Discovery (Basis for Supporting Innovative Drug Discovery and Life Science Research (BINDS)) from AMED under Grant Number JP22ama121001 (*support number 3118*). The synchrotron radiation experiments were performed at the BL32XU of SPring-8 with the approval of the Japan Synchrotron Radiation Research Institute (JASRI) (Proposal No. 2021A6700). We thank Kunio Hirata, Hiroaki Matsuura, and BL32XU beamline staff for assisting with X-ray crystallographic data collection and analysis. We are grateful to Naohiro Matsugaki for the diffraction data analysis, Yoshitaka Moriwaki for the AlphaFold2 analysis, and Yukishige Ito for the NMR analysis. We thank Yasuhiro Shimane, Miho Hirai, and Fumie Kondo for their assistance in this study. We thank Prof. Bernard Henrissat, Dr. Nicolas Terrapon and the CAZy teams for the family division of the four groups of enzymes. This work was supported by the Japan Society for the Promotion of Science [Grant Numbers JP20K15444 (S.H.), JP22K05398 (T. Sumida)) and the Mizutani Foundation for Glycoscience grant (T. Sumida). K.A.S. appreciates the support of the Australian Research Council (FT100100291).

## Author contributions

T. Sumida conceived and designed the study; performed molecular experiments, protein purifications, enzyme characterization, and protein crystallization; determined the protein structures; and wrote the manuscript. S.H. performed the bioinformatics analyses and wrote the manuscript. K.U. performed the molecular experiments and protein purifications. A.I. and K.T. performed NMR analysis. T. Sengoku performed structural prediction. K.A.S. synthesized inhibitors. S.D. and T.N. wrote the manuscript and supervised the project. S.F. determined the protein structures, wrote the manuscript, and supervised the project. All authors reviewed the manuscript draft and approved the final manuscript.

## Competing interests

The authors declare no competing interests.
