## [Peer Review File · Nature Communications]

Genetic and functional diversity of β -*N*-acetylgalactosamine
-targeting glycosidases expanded by deep-sea metagenome
analysisREVIEWER COMMENTS

Reviewer #1 (Remarks to the Author):

In this manuscript the authors present the discovery and characterization of eight novel β -N-acetylgalactosaminidases and β -N-acetylhexosaminidase using sequence-based screening and search of metagenomic databases. This is followed by the author's effort to characterize the new enzymes with experiments related to substrate specificity, biochemical properties of the enzymes, X-ray crystal structures, endo-exo specificity of the enzymes as well as their mechanism of action. The manuscript successfully established the importance of investigating NGAs in natural processes and provides valuable input to the research community with the discovery of these new enzymes. These are noteworthy and original results as they represent a large jump in the number of β -N-acetylgalactosaminidases and β -N-acetylhexosaminidases, which are biotechnologically useful enzymes.

There are a few points that they need to be addressed before publication. Also, a few errors were detected, and some parameters are missing from the methods and supporting information that would make the results easier to follow and reproduce.

Major comments

Table 1. The significant digits reported for the error of values in table 1 are too precise. For example, the authors report a value of 388.5 +/- 41.6 for the kcat of NgaB1. The error associated (41.6) cannot be determined this accurately, and should only be reported with the first digit (i.e. 390 +/- 40).

Page 8, line 214: No natural substrates were found for NgaCa, NgaMg or NgaDssm, so can you really support the theory about the exo-endo specificity of these enzymes based on the rest of the results? The statements regarding exo-/endo-activity should be confined to enzymes that have confirmed substrates.

Page 9, line 270: Without any actual substrate discovered for NgaCa, does the presence of certain domains in its' structure, provide enough evidence to support the endo-type beta-NGA activity? Could there also be conformational changes associated with substrate binding that change the active site geometry, and therefore change the predicted endo/exo-activity?

Page 17, line 533: There are not enough information about this experiment in the methods section. Did you use normal phase silica or C18? Did you use TLC-MS to check the mass of each spot after you run the TLC. Also, TLC is still an experiment so usually the plates used, solvents, reagents and commercially available compounds used should be mentioned in terms of purity and where they were purchased from.

Page 31, line 776, Table 1: What are the units for the relative activity of the enzymes? Also did you include any controls in these experiments? Perhaps some known NGAs and their substrates would be a good comparison to validate the importance of these results.

Page 36, line 808, Extended data Fig 5: I believe LC-MS experiments would be much more accurate to confirm specific hydrolysis of each oligosaccharide used. Unless there is a mass confirmation for the different products you are mentioning, the Rf values are enough to support these statements.

Page 42, line 856, Extended data Fig.8: A section about the NMR experiments should be included in the methods or the supporting information. There the equipment used should be mentioned as well as the software used to analyze the results. Additionally, integration of the peaks would help to follow the data

presented.

Minor comments

Page 7, lines 203-204: Referring to rapid or slow digestion based on TLC results is not standard. TLC on its own does not provide any time dependent information. Did you repeat these TLC in multiple time points to justify this statement?

Page 12, lines 369-370: The size of letters is smaller.

Page 14, line 428: What types of disease diagnosis are you referring to? Suitable references should be added to this paragraph.

Page 17, line 526: What is the buffer solution used? If it is the buffers listed in table 1, please refer to that here.

Page 43, lines 870-871: It is possible to suppress the water peak using presat settings on the NMR. Also why do the water peaks seem to be shifted in all spectra even though they are aligned to show the same area (in ppm)?

Page 44, line 873, Extended data Fig.9: Every 2D NMR should include two spectra on the y,x-axis. In the case of COSY, that means ¹H-NMR on the y-axis and ¹H-NMR on the x-axis. In this case the proton NMR is missing on the y-axis. That makes it difficult for the reader to interpret the data presented and the proton-proton interactions within the molecule. Also, this is just called a COSY spectra, not a COSY/¹H-NMR. Finally, the spectra are too small to follow properly and consider to use the same format for all spectra presented. That means, same color, presence or not of crosshairs, etc.

Reviewer #2 (Remarks to the Author):

The manuscript about the 'Genetic and functional diversity of β -N-acetylgalactosamine targeting glycosidases expanded by deep-sea metagenome analysis' by Tomomi Sumida et al. describes an extensive phylogenetic, biochemical and structural characterization of a diverse enzyme class, showing hydrolytic activity on β -N-acetylgalactosamine containing glycans. A conserved DUF domain, present in the sequences of identified enzymes of a meta-genomic dataset, was used for further data mining to explore the complete picture of phylogenetic and biochemical diversity of this novel class of enzymes. The sequences are related to enzymes from family GH123, but phylogenetically fall into four clades/groups that are indicative of divergence of substrate specificity and/or mode of action. The biochemical of eight and structural characterization of five representative enzymes from the four major groups underscore the diversification of substrate specificities.

The findings are new, the experiments are well conducted, and methods are well documented. Overall, the manuscript is well written, but I am a bit concerned about the nomenclature using the term of "endo"-acting enzymes without having measured activity on polysaccharide substrate. Activity on short oligosaccharides by cleaving of di-saccharides at the non-reducing-end have been reported for several other GH families, such as cellobio-hydrolases (Phylogenetic analysis of family 6 glycoside hydrolases. Mertz B, Kuczynski RS, Larsen RT, Hill AD, Reilly PJ. Biopolymers. 2005 Nov;79(4):197-206. doi: 10.1002/bip.20347) or chitobio-hydrolases (Characterization of a novel exo-chitosanase, an exo-

chitobiohydrolase, from *Gongronella butleri*. *J Biosci Bioeng* . 2019 Apr;127(4):425-429. doi: 10.1016/j.jbiosc.2018.09.009). These enzymes are not generally classified into endo-acting enzymes, a definition which rather concerns those that are active on polysaccharides, producing oligosaccharides of variable lengths ranging from DP2 to DP6 or even up to DP8. Since the described experiments only include assays on artificial short oligosaccharides or on globosides but not on polysaccharide substrate (which, I agree, are difficult to set up since such a polysaccharide is not commercially available), and also because none of the so called “endo”- active enzymes appears to be active on the Globo-H hexaose or the pentaose (Extended figures 5b,c, and d), I recommend to change the description of endo-acting enzymes to N-acetylgalactosidases or at least provide a short and precise definition of what the authors call “endo-acting”. Nevertheless, in my opinion, the study is within the scope of the journal, it contains largely original and significant results, and after changing or clarifying the activity designation throughout the manuscript, I only have detected some minor corrections that should be taken into account.

Comments

1. The authors have provided a tremendous amount of work, and I am not begging for more experiments, but why no attempt was made to express and characterize (an) enzyme(s) belonging to *dssm_1*?
2. Results Page 5 line 118-117. In my opinion the experiments do not allow to affirm that *NgaDssm* is endo-acting. What the experiments show is that the enzyme is capable of cleaving artificial substrates placing the non-natural aglycon pNP in the aglycone +1 site and the necessity to have a β -linked GalNAc positioned at -1 with respect to the cleavage site. It also demonstrates that the enzyme can accommodate/be tolerant for an additional sugar-unit at the -2 sub-binding site, but it could as well be (just) able to cleave mono and di-saccharides from the non-reducing end of short oligosaccharides (in Extended Figure 5b, c and d the maximum sized oligosaccharide that is degraded is DP4, a tetraose); other enzymes of this type have been reported (see references mentioned in my global assessment above) – but it does not (yet) mean that it is an overall endo-acting enzyme – this would need experimental demonstration by cleavage of a longer β -GalNAc oligo- (at least DP6) or even polysaccharide.
3. Results Page 5 line 122-131. It does not become clear from the text whether the sequences of type *dssm_1* have been searched for and included in the phylogeny. For example, from the color coding of the predicted structures in Figure 1a, it is not clear if *dssm_1* also contains the DUF domain or not. Also, the correspondence of *dssm_1*, *dssm_2* and *dssm_3* and groups 1 to 3 of the phylogeny are unclear and thus should be more explicitly defined. Please provide more details for more precision.
4. Results page 5 line 133 “..among all three domains of life, 14 representative sequences of each novel group were selected...”
The reason why 14 sequences were selected for the phylogeny should be explained in the corresponding methods section, page 15 section “bioinformatics” lines 459 and following.
5. Discussion page 13, lines 388 and 389. “In the CAZy database, several GH families are grouped into “clans” according to the conservation of catalytic residues, ...”. Why do the authors believe that the

hierarchical level corresponds to a “clan”? Defining these groups of enzymes as a “clan” would mean that the authors believe to be in the presence of three new GH families (corresponding to groups 1, 2 and 3), which are distantly related to GH123. One could also consider these enzymes to create distinct subfamilies **within** GH123, in view of the non-negligible similarity among the groups, at least at the catalytic core (i.e. superimposition on figure 2). Indeed, although low, 15-20% identity allows to align these enzymes (as shown in figure 1), and the overall common fold and significant features such as the presence of the DUF4091-domain in all, might be the arguments to rather plead for the latter, these being subfamilies of GH123; but this definitively needs to be discussed with the CAZY-team.

6. Supplemental Tables 1 to 5. In all the tables reporting the crystallographic data parameters and statistics the authors should reduce the number of significant digits: all unit cell parameter values and R-factors should only be given with 2 digits after the comma; in general, X-ray measurements do not allow to be more precise than that.

Reviewer #3 (Remarks to the Author):

This manuscript showcases an interesting enzyme discovery effort to find GalNAc-specific enzymes from deep sea metagenomic samples and then broader to other known organisms. The manuscript is well-written and for the most part easy to follow. The potential discovery of three new CAZY families is significant as is the suggestion of forming a new clan within CAZY, and the structural biology is a very nice addition. Also, successfully using a smaller DUF within a larger enzyme as search query is a nice strategy that could be useful to other researchers focusing on enzyme discovery.

Comments:

Firstly, I think the authors should reach out to CAZY to determine whether these identified groups are in fact new CAZY families. The manuscript can then be updated with GHXXX, GHYYY, GHZZZ, or similar and the new family names be ready to launch upon possible acceptance later on. The same goes for the suggested clan. (the CAZY crew does not have to become authors for this preparation necessarily, to my knowledge)

My main concern with the manuscript is the description of the enzymology, which I believe requires further explanation or possibly it needs to be repeated to be conclusive. The reason for choosing 30-min assays for pNP substrates is not clear, because this is a stopped assay. Did you verify that this represents an initial/linear rate of the reaction, and if so, how? If not, the rates monitored are not useful.

What substrates and concentrations were used for pH optimum determination? How was optimal temperature determined? This is not clear from the methods. Optimal temperature is also somewhat subjective/not strictly defined, as it depends also on the length of the reaction whether the enzyme is working optimally and the T giving the fastest reaction rate is often close also to the thermal denaturation point, so this needs to be explained.

It does not appear that you have reached saturation in your kinetic assays, at least for some reactions

from what is visible in Extended data Fig. 3., which makes the kinetic parameters untrustworthy, and they can only be described as apparent right now. You need to increase the substrate concentration, or alternatively only calculate k_{cat}/K_m using linear regression at low $[S]$.

I think the classification of endo-acting enzymes is a bit weak when you only have small saccharides as substrates and not longer chains, as cleaving two sugars in from a chain is not obviously exo but also not clearly an activity that cleaves along an extended chain. It would be good to specify this limitation and that tests on longer polysaccharides could provide a more conclusive endo/exo behavior.

How were enzyme subsites defined? You don't have longer ligands in your structures, so the designation seems somewhat arbitrary. Please clarify this in the text.

I also think the authors should explain why they would look for GalNAc-active enzymes in the deep sea. The rationale for the study is quite vaguely described, for instance on lines 67-70, a need to find new β -NAGs based on little being known currently is in my opinion only a starting point to the aim of the study – why do we need more information? What is, or could be, their relevance? More concrete examples of GalNAc containing glycans would be helpful to put the study into a broader context, and I think showing these in a figure would be helpful for the reader, which could be done with symbol nomenclature or actual chemical structures. In connection, the reason to look for these enzymes in the deep sea is somewhat unclear. Why was this environment chosen? Clearly you discovered terrestrial enzymes afterwards through bioinformatic means, so what could be the roles of the enzymes in a deep-sea environment compared to in e.g. a plant like Arabidopsis? Stating that “the deep-sea microbiome is an attractive potential bioresource for screening undiscovered enzymes”, lines 370-371, is still vague in terms of the activity you searched for, which clearly was not limited to that environment... I understand if you chose this environment because it was “cool”, but please rephrase it to better frame the story of the manuscript.

The possible biological roles of these enzymes could very well be discussed further, to give some more context as to what they might be active on. For example, on lines 270-271 you state that “This structural feature indicated that Group 1 enzymes act on the inner regions of longer sugar chains”. Which types of longer sugar chains are those? From the STRING database analysis, it seems like many of the enzymes could have activities related to exopolysaccharide remodeling, since you have a mixture of transporters, degradative, and synthesizing enzymes, which could be expanded by giving some more examples of how these structures may look. In connection to this, I think it would be helpful to show the structures of the substrates tested, either in the main text or in the supplementary information (either using symbol nomenclature or actual chemical structures), as the extended data will not be as accessible to the reader.

We express our gratitude to the reviewers for their positive and constructive feedback on our manuscript. The insightful comments provided were invaluable, and consequently, the manuscript has been thoroughly revised and improved.

In response to comments from reviewers and the editor, we have performed additional experiments to investigate the substrate specificity of the enzymes referred to as “endo-types” in the previous manuscript. We performed experiments with longer oligosaccharides (GM1b: NeuAc α 2-3Gal β 1-3GalNAc β 1-4Gal β 1-4-Glu and SSEA-4: NeuAc α 2-3Gal β 1-3GalNAc β 1-3Gal α 1-4Gal β 1-4Glu) and chondroitin sulfate (A, B, and C), a polymer containing (4 and/or 6)-sulfated GalNAc, and found that none of the endo-type enzymes acted on chondroitin sulfates (Supplementary Fig. 7, which has been newly added in this revision); however, NgaBa exhibited activity on GM1b and SSEA-4 oligosaccharides to release trisaccharides from their non-reducing ends (Fig. 3f, g, which has been newly added in this revision). However, tests with even longer oligosaccharides, which are critical for the final classification of end-acting enzymes, were not feasible because of the unavailability of commercially accessible substrates. Therefore, we have decided to use the terms “oligosaccharide-releasing (OR)” and “monosaccharide-releasing (MR)” to classify the substrate specificity of our enzymes, instead of the terms “endo-type” and “exo-type”. Moreover, considering the suggestions provided by CAZy teams during the revision process, we have revised the group names of the newly identified enzymes from Group 1, Group 2, Group 3-1, and Group 3-2 in the previous manuscript to Group 1, Group 2, Group 3, and Group 4, respectively, in this manuscript.

We have reduced the number of display items to less than 10, in line with the formatting guidelines of *Nature Communications*, while moving additional Figures, Tables, and Data to Supplementary Information. Supplementary Fig. 1, 2c–e, 5, 6, 7, 13a, 19 have been newly incorporated in this revision.

Our point-by-point response to the reviewer’s comments is presented below.

Response to Reviewer #1's comments

In this manuscript the authors present the discovery and characterization of eight novel β -N-acetylgalactosaminidases and β -N-acetylhexosaminidase using sequence-based screening and search of metagenomic databases. This is followed by the author's effort to characterize the new enzymes with experiments related to substrate specificity, biochemical properties of the enzymes, X-ray crystal structures, endo-exo specificity of the enzymes as well as their mechanism of action. The manuscript successfully established the importance of investigating NGAs in natural processes and provides valuable input to the research community with the discovery of these new enzymes. These are noteworthy and original results as they represent a large jump in the number of β -N-acetylgalactosaminidases and β -N-acetylhexosaminidases, which are biotechnologically useful enzymes.

There are a few points that they need to be addressed before publication. Also, a few errors were detected, and some parameters are missing from the methods and supporting information that would make the results easier to follow and reproduce.

Major comments

1. Table 1. The significant digits reported for the error of values in table 1 are too precise. For example, the authors report a value of 388.5 +/- 41.6 for the *kat* of NgaB1. The error associated (41.6) cannot be determined this accurately, and should only be reported with the first digit (i.e. 390 +/- 40).

→ Thank you for your suggestion. We have revised the significant digits per your suggestions.

	NgaDssm	NgaCa	NgaMg	NgaAt	NgaBa	NgaBf	NgaBl	NgaLy	NgaP2	NgaP*
	Group 3	Group 1	Group 1	Group 2	Group 3	Group 4	Group 4	Group 4	Group 4	GH123
Substrate specificity										
Substrate	Relative activity (%)									
GalNAc- β - p NP	100	-	-	100	100	100	100	100	100	100
Gal β 1-3GalNAc- β - p NP	70.3	100	-	-	7.3	-	-	-	-	NT
Gal- β - p NP	-	-	-	-	-	-	-	-	-	-
GlcNAc- β - p NP	-	-	-	13.6	-	-	-	-	4.7	-
GalNAc- α - p NP	-	-	-	-	-	-	-	-	-	-
Substrate	Relative activity (%)									
GalNAc- β -4MU	100	-	-	100	100	100	100	100	100	100
GalNAc4S- β -4MU	183.8	-	-	47.9	1.5	1.9	2.0	1.3	29.0	-
GalNAc6S- β -4MU	-	-	-	-	-	-	-	-	-	-
General property										
Enzyme activity	OR/MR- β -NGA	OR- β -NGA	unknown	OR/MR- β -HEX	OR/MR- β -NGA	MR- β -NGA				MR- β -NGA
Optimal pH	5.0	5.0	-	5.0	6.5	6.5	5.0	7.0	5.5	6.0
Optimal buffer	Citrate	Citrate	-	Citrate	Citrate	Citrate	Citrate	HEPES	Citrate	Acetate
Optimal temperature (°C)	45	25	-	40	70	35	45	40	40	NT
T_m (°C)	63.9	43.1	70.6	52.9	93.1	44.3	57.2	59.7	56.8	NT

K_m (mM)	(3.8±0.4)	(11±3)	-	(12±5)	(3.9±1)	1.0±0.07	(2.8±0.4)	1.0±0.05	0.36±0.03	0.4
k_{cat} (s ⁻¹)	(43±3)	(1.2±0.3)	-	(8.0±3)	(510±50)	170±5	(390±40)	480±10	28±0.6	7.3
k_{cat}/K_m (s ⁻¹ mM ⁻¹)	(11)	(0.11)	-	(0.64)	(130)	170	(140)	490	79	21

2. Page 8, line 214: No natural substrates were found for NgaCa, NgaMg, or NgaDssm, so can you really support the theory about the exo-endo specificity of these enzymes based on the rest of the results? The statements regarding exo-/endo-activity should be confined to enzymes that have confirmed substrates.

→ In response to feedback provided by yourself and the other reviewers, we have modified the terms “endo-” and “exo-” activity to “oligosaccharide-releasing (OR)-” and “monosaccharide-releasing (MR)-” activity, respectively (please see our general, opening response above). As NgaCa and NgaDssm acted on Galβ1-3GalNAc-β-pNP, they are defined as oligosaccharide-releasing β-NGA (OR-β-NGA) in the revised manuscript. The NgaMg activity is reported as “unknown” in Table 1 because NgaMg exhibited no activity on any of the tested substrates.

3. Page 9, line 270: Without any actual substrate discovered for NgaCa, does the presence of certain domains in its structure, provide enough evidence to support the endo-type beta-NGA activity? Could there also be conformational changes associated with substrate binding that change the active site geometry, and therefore change the predicted endo/exo-activity?

→ NgaCa might be involved in regulating the lipoproteins as shown in Supplementary Fig. 18 and Supplementary Data. S4 and S5 (page 15–16, lines 467–472). Additionally, the ganglio- and globo-series substrates used in this study may not represent the actual target substrates of NgaCa. GalNAc-containing oligosaccharides exhibit diverse structures (Supplementary Fig. 1, which has been newly added in this revision). Unfortunately, only a limited number of substrate varieties are commercially available. Therefore, performing a more in-depth examination of the substrate is challenging, and the actual substrate for NgaCa (or NgaDssm, NgaMg) could not be conclusively identified in this study.

NgaCa lacks any unique substructure that represents monosaccharide-releasing (MR) enzymes, such as an additional loop (Fig. 6d and 6e, colored blue and magenta). Although a conformational change between open and close conformations was observed in the movement of the loop near the cleft structure (Supplementary Fig. 13c), this was not a conformational change that would change the cleft structure of the substrate binding site. In addition, NgaCa lacks any loops or additional domains that would cause a conformational change from a cleft structure to a pocket structure upon substrate binding. Consequently, NgaCa would not have the potential for MR/OR alterations.

The substrate binding site of NgaCa exhibits a cleft structure; in addition, three characteristic tryptophan residues (W197, W218, W249) are located in the subsites -2 and -3 (Supplementary Fig. 13a, which has been newly added in this revision). Such a cleft structure and the presence of aromatic amino acids (such as tryptophan or phenylalanine) in the substrate binding site are characteristically observed in endo-type enzymes, such as chitinase or xyloglucanase, which bind long oligosaccharides (references below; Kurašin et al., Ariza et al., Matsuzawa et al.). A docking model of NgaCa with oligosaccharide suggested that these tryptophan residues may play a role in a stacking-type interaction with sugars located in subsites -2 and -3. This structural feature indicates that Group 1 enzymes may act on the inner regions of longer sugar chains, in addition to disaccharides from the non-reducing end. These points are elaborated in the Result section (Pages 11, Lines 309–315).

<References>

[Kurašin et al. described that “The long chitin binding cleft is lined with aromatic residues, mostly Trp and Phe (Fig. 1). The hydrophobic interactions form a flexible sheath necessary for substrate recognition and sliding of the polymer chain between successive glycosidic bond cleavages.” (JBC, 290, 29074-85, 2015)].

[Ariza et al. described that “Substrate recognition is achieved by the interactions between aromatic residues and pyranose rings and several hydrogen bonds with hydroxyl groups of pyranose rings. Around the catalytic center, subsites -1, -2, and -3 form hydrogen bonds, and subsites -1, -3, and -4 form aromatic stacking with substrates.” (JBC, 286, 33890-33990, 2011)]

[Matsuzawa et al. described that “*Paenibacillus* endo-xyloglucanase has four characteristic tryptophan residues (W61, W64, W318, and W319) around the active site cleft. W61 and W64 are found in the “negative” subsites. In contrast, W318 and W319 are found in the “positive” subsites. Exo-type enzymes, such as *Geotrichum* OXG-RCBH and *Aspergillus* OREX, lack tryptophan residues analogous to W61, W64, W318, and W319”. (FEBS Letters, 588, 1731-38, 2014)].

4. Page 17, line 533: There are not enough information about this experiment in the methods section. Did you use normal phase silica or C18? Did you use TLC-MS to check the mass of each spot after you run the TLC. Also, TLC is still an experiment so usually the plates used, solvents, reagents and commercially available compounds used should be mentioned in terms of purity and where they were purchased from.

→ Thank you for pointing this out. We used a precoated Silica Gel 60 TLC plate for the experiment. Detailed information about the TLC experimental setup (such as information on the plate used,

solvents, reagents, commercially available compounds, etc.) has been added to the Methods section. (Pages 20–21). We could not perform mass spectrometry for each spot using TLC-MS and LC-MS as mentioned in your comment. However, in this revised version, we have added the R_f values of each spot in Fig. 3a. Owing to the challenges of detecting the non-reducing terminal trisaccharides of SSEA-4 and GM1b (which have been included as additional experiments in this revision) using the diphenylamine-aniline-phosphate reagent, we used resorcinol-HCl reagents, specifically designed for the detection of sialic acid-containing oligosaccharides. (Figure 3g and Supplementary Fig. 6c, which have been newly added in this revision). Details have been incorporated into the Methods section (Page 20–21, lines 633–638).

5. *Page 31, line 776, Table 1: What are the units for the relative activity of the enzymes? Also, did you include any controls in these experiments? Perhaps some known NGAs and their substates would be a good comparison to validate the importance of these results.*

→ We have added “Units (%)” to Table 1. Negative controls were measured using a buffer solution without the enzyme as blank, and the relative activity was calculated as 100% of the activity of GalNAc-β-pNP (or Galβ1-3GalNAc-β-pNP for NgaCa) or GalNAc-β-4MU, which are now mentioned in the legend of Table 1. The previously reported values for GH123 NgaP have been included to the last column on the right in Table 1.

6. *Page 36, line 808, Extended data Fig 5: I believe LC-MS experiments would be much more accurate to confirm specific hydrolysis of each oligosaccharide used. Unless there is a mass confirmation for the different products you are mentioning, the R_f values are enough to support these statements.*

→ We agree that the LC-MS measurements are much more accurate; however, regrettably, we lack the experimental facilities required to perform such measurements. As an alternative, we have added the R_f values for each band in Fig. 3a. The Methods section now specifies that the reaction products were identified by comparing their R_f values with those of the standard oligosaccharides and sugar (Page 21, lanes 638–639).

7. *Page 42, line 856, Extended data Fig.8: A section about the NMR experiments should be included in the methods or the supporting information. There the equipment used should be mentioned as well as the software used to analyze the results. Additionally, integration of the peaks would help to follow the data presented.*

→ We have added information on the NMR experiments to the Methods section (Page 22) as per your suggestion. Additionally, we have added the integrations of the peaks of the hydrolysates after equilibration (Fig. 8a, lane 5). The integrations of the peaks of the substrates are reported in Supplementary figures and tables.

Minor comments

1. *Page 7, lines 203-204: Referring to rapid or slow digestion based on TLC results is not standard. TLC on its own does not provide any time dependent information. Did you repeat these TLC in multiple time points to justify this statement?*

→ Degradation experiments using TLC were performed at reaction times of 4 h and 16 h. Although consistent results were obtained at both time points, we have revised the term “rapid digestion”. (Page 8, line 233)

2. *Page 12, lines 369-370: The size of letters is smaller.*

→ Thank you for pointing this out and for carefully reading the text. We have now corrected the size of the letters accordingly.

3. *Page 14, line 428: What types of disease diagnosis are you referring to? Suitable references should be added to this paragraph.*

→ GH123 β -NGA has been employed in newborn screening for mucopolysaccharidosis, a congenital metabolic disorder. The citation for this article has been included at the relevant instance in the revised text (Page 16, line 493).

4. *Page 17, line 526: What is the buffer solution used? If it is the buffers listed in table 1, please refer to that here.*

→ The buffers used in our experiments have been listed in Table 1, and this information has been added in the manuscript (Page 20).

5. *Page 43, lines 870-871: It is possible to suppress the water peak using presat settings on the NMR. Also why do the water peaks seem to be shifted in all spectra even though they are aligned to show the same area (in ppm)?*

→ As shown in the figure, we used presaturation in the experiment. Owing to constraints related to limited quantities of enzymes available for the NMR experiment and uncertainties about enzyme stability during solvent exchange to D₂O, we exclusively used the enzyme solution in H₂O with buffering materials, which were also evident as prominent peaks in the ¹H NMR chart. As the peaks of products overlapped with or were very close to the large HOD area, we could not acquire well-defined pre-saturated spectra. Fortunately, we were able to observe the desired peaks in the ¹H NMR chart of enzymatically cleaved products of the substrate, facilitating the assignment of the initial hydrolysis product, as previously demonstrated.

As pointed out by you, the water peaks appear to be shifted in all spectra, primarily owing to temperature differences. We had previously mentioned in the caption of the NMR monitoring this. Specifically, the HOD peaks are pre-saturated at 4.8 for measurements at room temperature and at 4.63 for monitoring at 37°C.

6. *Page 44, line 873, Extended data Fig.9: Every 2D NMR should include two spectra on the y,x-axis. In the case of COSY, that means 1H-NMR on the y-axis and 1H-NMR on the x-axis. In this case the proton NMR is missing on the y-axis. That makes it difficult for the reader to interpret the data presented and the proton-proton interactions within the molecule. Also, this is just called a COSY spectra, not a COSY/1H-NMR. Finally, the spectra are too small to follow properly and consider to use the same format for all spectra presented. That means, same color, presence or not of crosshairs, etc.*

→ We have revised the charts as per your recommendations. Additionally, we have expanded Fig. 8 (Extended data Fig. 9 in the previous manuscript) to the best possible extent. We believe readers can further enlarge it on their PCs if necessary. Also, instead of calling COSY spectra, we used 2D ¹H-¹H COSY spectrum.

However, in the case of Supplementary Fig. 15, we could not add the ¹³C on the y-axis because the amount of the sample used for NMR monitoring is too small, making it impractical to obtain ¹³C NMR spectra. The HMQC was employed for peak reading, and assignment of ¹³C NMR of the products.

Response to Reviewer #2's comments

The manuscript about the 'Genetic and functional diversity of β -N-acetylgalactosamine targeting glycosidases expanded by deep-sea metagenome analysis' by Tomomi Sumida et al. describes an extensive phylogenetic, biochemical and structural characterization of a diverse enzyme class, showing hydrolytic activity on β -N-acetylgalactosamine containing glycans. A conserved DUF domain, present in the sequences of identified enzymes of a meta-genomic dataset, was used for further data mining to explore the complete picture of phylogenetic and biochemical diversity of this novel class of enzymes. The sequences are related to enzymes from family GH123, but phylogenetically fall into four clades/groups that are indicative of divergence of substrate specificity and/or mode of action. The biochemical and structural characterization of five representative enzymes from the four major groups underscore the diversification of substrate specificities.

The findings are new, the experiments are well conducted, and methods are well documented. Overall, the manuscript is well written, but I am a bit concerned about the nomenclature using the term of "endo"-acting enzymes without having measured activity on polysaccharide substrate. Activity on short oligosaccharides by cleaving of di-saccharides at the non-reducing-end have been reported for several other GH families, such as cellobio-hydrolases (Phylogenetic analysis of family 6 glycoside hydrolases. Mertz B, Kuczynski RS, Larsen RT, Hill AD, Reilly PJ. Biopolymers. 2005 Nov;79(4):197-206. doi: 10.1002/bip.20347) or chitobio-hydrolases (Characterization of a novel exo-chitosanase, an exo-chitobiohydrolase, from Gongronella butleri. J Biosci Bioeng. 2019 Apr;127(4):425-429. doi: 10.1016/j.jbiosc.2018.09.009). These enzymes are not generally classified into endo-acting enzymes, a definition which rather concerns those that are active on polysaccharides, producing oligosaccharides of variable lengths ranging from DP2 to DP6 or even up to DP8. Since the described experiments only include assays on artificial short oligosaccharides or on globosides but not on polysaccharide substrate (which, I agree, are difficult to set up since such a polysaccharide is not commercially available), and also because none of the so called "endo"-active enzymes appears to be active on the Globo-H hexaose or the pentaose (Extended figures 5b,c, and d), I recommend to change the description of endo-acting enzymes to N-acetylgalactobiosidases or at least provide a short and precise definition of what the authors call "endo-acting". Nevertheless, in my opinion, the study is within the scope of the journal, it contains largely original and significant results, and after changing or clarifying the activity designation throughout the manuscript, I only have detected some minor corrections that should be taken into account.

Comments

1. *The authors have provided a tremendous amount of work, and I am not begging for more experiments, but why no attempt was made to express and characterize (an) enzyme(s) belonging to dssm_1?*

→ We did not express and characterize DSSM_1 because it was not identified as an enzyme that acts on β -GalNAc. Notably, dssm_1 showed sequence similarity in the N-terminal domain 2 (amino acids 239–379 in DSSM_1) to GH123. However, the putative catalytic domain (domain 4 in DSSM_1) was not conserved with GH123, and neither the DE site nor DUF4091 was present. Thus, this enzyme is likely a false positive resulting from the sequence-based search and is not expected to function as an enzyme acting on β -GalNAc. Thus, dssm_1 was not selected for the subsequent enzymatic experiments in this study. We have included additional explanations in the Result section as follows: DSSM_1 was structurally distinct from the GH123 β -NGAs (Fig. 1a, b) and comprised five domains, among which only domain 2 (β -sandwich) displayed a degree of structural similarity to the N-terminal domain of GH123 β -NGAs. Domain 4 ($(\beta/\alpha)_8$ barrel) was similar to GH66 cycloisomaltooligosaccharide glucanotransferase (PDB, 3WNM), although a blast search using the dssm_1 sequence did not yield significant similarity with enzymes registered in the PDB. Based on its structural characteristics, dssm_1 was not considered as a candidate sequence for β -NGA (Page 5, lines 115–121).

2. *Results Page 5 line 118-117. In my opinion the experiments do not allow to affirm that NgaDssm is endo-acting. What the experiments show is that the enzyme is capable of cleaving artificial substrates placing the non-natural aglycon pNP in the aglycone +1 site and the necessity to have a β -linked GalNAc positioned at -1 with respect to the cleavage site. It also demonstrates that the enzyme can accommodate/be tolerant for an additional sugar-unit at the -2 sub-binding site, but it could as well be (just) able to cleave mono and di-saccharides from the non-reducing end of short oligosaccharides (in Extended Figure 5b, c and d the maximum sized oligosaccharide that is degraded is DP4, a tetraose); other enzymes of this type have been reported (see references mentioned in my global assessment above) – but it does not (yet) mean that it is an overall endo-acting enzyme – this would need experimental demonstration by cleavage of a longer β -GalNAc oligo- (at least DP6) or even poly-saccharide.*

→ Thank you for these valuable comments. NgaDssm was active against Gal β 1-3GalNAc- β -pNP and GalNAc- β -pNP but not against natural substrate. In this study, however, we found that NgaBa released disaccharides from the non-reducing end of DP5 oligosaccharides (Gb5) [Fig. 3f (Extended Data Fig. 5d in the previous manuscript)]. In addition, in this revision, we have included data obtained

from experiments involving longer β -GalNAc-containing oligosaccharides SSEA-4 (DP6, a hexaose) and GM1b (DP5, a pentaose) as substrates and found that NgaBa released trisaccharides from the non-reducing ends of SSEA-4 and GM1b (Figure 3f, g and Supplementary Fig. 6b, c, which are newly added in this revision). However, as you have rightly pointed out, experiments using polysaccharides as substrates are essential to confirm novel β -NGAs as an overall endo-type enzyme. However, owing to the unavailability of such substrates, we have opted to use the term “oligosaccharide-releasing enzymes” instead of “endo-acting enzymes.” (please see our general, opening response above)

3. *Results Page 5 line 122-131. It does not become clear from the text whether the sequences of type dssm_1 have been searched for and included in the phylogeny. For example, from the color coding of the predicted structures in Figure 1a, it is not clear if dssm_1 also contains the DUF domain or not. Also, the correspondence of dssm_1, dssm_2 and dssm_3 and groups 1 to 3 of the phylogeny are unclear and thus should be more explicitly defined. Please provide more details for more precision.*

→ Thank you for pointing this out. The phylogenetic tree was constructed using sequences that contained DUF4091. Dssm_1 was not included in the phylogenetic tree because it does not contain DUF4091. Here, we have indicated the nodes of dssm_2 and dssm_3 in Fig. 1c. All the sequence IDs used for the phylogenetic analysis and their corresponding groups have been provided in Supplementary Data S2. This point has been added to the legend of Figure 1. Additionally, we have emphasized the position of DUF4091 in the sequence alignment (Supplementary Fig. 2b).

4. *Results page 5 line 133 “..among all three domains of life, 14 representative sequences of each novel group were selected...”*

The reason why 14 sequences were selected for the phylogeny should be explained in the corresponding methods section, page 15 section “bioinformatics” lines 459 and following.

→ All previously reported GH123 enzymes were exclusively derived from specific bacteria lineages (i. e., Bacillota (NgaP and CpNga123) and Bacteroidota (BvGH123)). In this study, to comprehensively examine diverse β -NGAs in nature and enhance our understanding of the enzyme, we manually selected 14 sequences from a range of diverse organisms (plants, archaea, and various bacterial phyla) for a thorough coverage of the four Groups defined based on phylogeny (Figure 1c). We have added explanations in Page 6, lines 154–159. Furthermore, we have omitted the word “representative” to prevent any potential misinterpretation.

5. *Discussion page 13, lines 388 and 389. “In the CAZy database, several GH families are grouped into “clans” according to the conservation of catalytic residues, ...”. Why do the authors believe that the hierarchical level corresponds to a “clan”? Defining these groups of enzymes as a “clan” would mean that the authors believe to be in the presence of three new GH families (corresponding to groups 1, 2 and 3), which are distantly related to GH123. One could also consider these enzymes to create distinct subfamilies within GH123, in view of the non-negligible similarity among the groups, at least at the catalytic core (i.e. superimposition on figure 2). Indeed, although low, 15-20% identity allows to align these enzymes (as shown in figure 1), and the overall common fold and significant features such as the presence of the DUF4091-domain in all, might be the arguments to rather plead for the latter, these being subfamilies of GH123; but this definitively needs to be discussed with the CAZY-team.*

→ Thank you for the suggestion. We have contacted the CAZY team. According to their suggestion as outlined below, we wish to modify the group names and definitions to subfamilies under the GH123 family. We have included descriptions about this in the Discussion section (Pages 14–15, lines 435–440).

“For the family division of the novel groups of enzymes identified in this study, the CAZY team’s bioinformatics perspective on the sequence was concluded as follows: there are subfamilies (Groups 3 and 4) closer to the main (sub)family, GH123, and distant groups (Groups 1 and 2). The distant subgroups (Groups 1 and 2) exhibit highly distinct sequence profiles could potentially be categorized as novel families. However, given their size and functional conservation, they were designated as a distant subfamily.”

6. *Supplemental Tables 1 to 5. In all the tables reporting the crystallographic data parameters and statistics the authors should reduce the number of significant digits: all unit cell parameter values and R-factors should only be given with 2 digits after the comma; in general, X-ray measurements do not allow to be more precise than that.*

→ We have reduced the number of significant digits for unit cell parameter values, in accordance with your suggestions. Three significant digits were used for the R-factor because it is a calculated value. Regarding the wavelength, we contacted the SPring-8 beamline staff for clarification. They responded as follows: “In accordance with the Si(111) energy resolution, the wavelength calibration is measured to 1E-5 digits and thus we used the values rounded to 1E-4 digits.”

Response to Reviewer #3's comments

This manuscript showcases an interesting enzyme discovery effort to find GalNAc-specific enzymes from deep sea metagenomic samples and then broader to other known organisms. The manuscript is well-written and for the most part easy to follow. The potential discovery of three new CAZy families is significant as is the suggestion of forming a new clan within CAZy, and the structural biology is a very nice addition. Also, successfully using a smaller DUF within a larger enzyme as search query is a nice strategy that could be useful to other researchers focusing on enzyme discovery.

Comments:

- 1. Firstly, I think the authors should reach out to CAZy to determine whether these identified groups are in fact new CAZy families. The manuscript can then be updated with GHXXX, GHYYY, GHZZZ, or similar and the new family names be ready to launch upon possible acceptance later on. The same goes for the suggested clan. (the CAZy crew does not have to become authors for this preparation necessarily, to my knowledge)*

→ Thank you for your kind suggestions. We have contacted the CAZy team and received suggestions that, although Groups 1 and 2 are phylogenetically highly distant from the previously reported GH123, it is reasonable to consider them as new subfamilies rather than new families. This is now mentioned in the Discussion section. (Pages 14–15, lines 435–442)

“For the family division of the novel groups of enzymes identified in this study, the CAZy team’s bioinformatics perspective on the sequence was concluded as follows: there are subfamilies (Groups 3 and 4) closer to the main (sub)family, GH123, and distant groups (Groups 1 and 2). The distant subgroups (Groups 1 and 2) exhibit highly distinct sequence profiles could potentially be categorized as novel families. However, given their size and functional conservation, they were designated as a distant subfamily. Following these decisions, GH123 was reconstituted as a large GH family with several subfamilies, comprising the newly identified enzyme groups (groups 1, 2, 3, and 4 and GH123).”

- 2. My main concern with the manuscript is the description of the enzymology, which I believe requires further explanation or possibly it needs to be repeated to be conclusive. The reason for choosing 30-min assays for pNP substrates is not clear, because this is a stopped assay. Did you verify that this*

represents an initial/linear rate of the reaction, and if so, how? If not, the rates monitored are not useful.

→ Thank you for pointing this out. We agree with this comment and have added the time course in Supplementary Fig. 19 (newly added in this revision). Additionally, in the Methods section, we have included that the time course indicates that the 30-min assays were measured at the initial rate (Supplementary Fig. 19) (Page 19, 587–588).

3. *What substrates and concentrations were used for pH optimum determination? How was optimal temperature determined? This is not clear from the methods. Optimal temperature is also somewhat subjective/not strictly defined, as it depends also on the length of the reaction whether the enzyme is working optimally and the T giving the fastest reaction rate is often close also to the thermal denaturation point, so this needs to be explained.*

→ We agree with the reviewer that the methods and explanations were inadequate. We have added more details about the reaction conditions and experimental methods in the Methods section (Pages 19–20). In addition, we have included an explanation of the relationship between the optimal temperature and T_m value in the Results section (Page 8, 215–217). “The melt curve plot suggested that the optimal temperature was very close to the thermal denaturation point, with the optimal temperature being approximately 10–20 °C lower than the T_m value (Fig. 2d, e).”

4. *It does not appear that you have reached saturation in your kinetic assays, at least for some reactions from what is visible in Extended data Fig. 3., which makes the kinetic parameters untrustworthy, and they can only be described as apparent right now. You need to increase the substrate concentration, or alternatively only calculate k_{cat}/K_m using linear regression at low [S].*

→ We agree with the reviewer that some reactions did not reach saturation in our experiments. This is because the substrate, GalNAc- β -pNP, did not dissolve in the buffer solution at concentrations above 3 mM; therefore, all experiments had to be performed with a maximum substrate concentration of 2 mM. For those enzymes that did not reach saturation, it is now clearly mentioned in the Result section that these values are apparent values because saturation was not reached. Table 1 shows these values in parentheses (Page 8, 219–223).

	NgaDssm	NgaCa	NgaMg	NgaAt	NgaBa	NgaBf	NgaBl	NgaLy	NgaP2	NgaP*
	Group 3	Group 1	Group 1	Group 2	Group 3	Group 4	Group 4	Group 4	Group 4	GH123
Substrate specificity										

Substrate	Relative activity (%)									
GalNAc- β -pNP	100	-	-	100	100	100	100	100	100	100
Gal β 1-3GalNAc- β -pNP	70.3	100	-	-	7.3	-	-	-	-	NT
Gal- β -pNP	-	-	-	-	-	-	-	-	-	-
GlcNAc- β -pNP	-	-	-	13.6	-	-	-	-	4.7	-
GalNAc- α -pNP	-	-	-	-	-	-	-	-	-	-
Substrate	Relative activity (%)									
GalNAc- β -4MU	100	-	-	100	100	100	100	100	100	100
GalNAc4S- β -4MU	183.8	-	-	47.9	1.5	1.9	2.0	1.3	29.0	-
GalNAc6S- β -4MU	-	-	-	-	-	-	-	-	-	-
General property										
Enzyme activity	OR/MR- β -NGA	OR- β -NGA	unknown	OR/MR- β -HEX	OR/MR- β -NGA	MR- β -NGA				MR- β -NGA
Optimal pH	5.0	5.0	-	5.0	6.5	6.5	5.0	7.0	5.5	6.0
Optimal buffer	Citrate	Citrate	-	Citrate	Citrate	Citrate	Citrate	HEPES	Citrate	Acetate
Optimal temperature (°C)	45	25	-	40	70	35	45	40	40	NT
T_m (°C)	63.9	43.1	70.6	52.9	93.1	44.3	57.2	59.7	56.8	NT
K_m (mM)	(3.8±0.4)	(11±3)	-	(12±5)	(3.9±1)	1.0±0.07	(2.8±0.4)	1.0±0.05	0.36±0.03	0.4
k_{cat} (s ⁻¹)	(43±3)	(1.2±0.3)	-	(8.0±3)	(510±50)	170±5	(390±40)	480±10	28±0.6	7.3
k_{cat}/K_m (s ⁻¹ mM ⁻¹)	(11)	(0.11)	-	(0.64)	(130)	170	(140)	490	79	21

-, activity<0.5%. Values are presented as means of technical triplicate experiments. The general properties of each β -NGA are listed based on the results in Extended Data Fig. 3. Controls were measured without the enzyme, serving as blanks, and the relative activity was calculated as 100% of the activity of GalNAc- β -pNP (or Gal β 1-3GalNAc- β -pNP for NgaCa) or GalNAc- β -4MU. The enzymes that did not reach saturation are indicated in parentheses in Table 1 owing to their apparent values. * NgaP values are listed based on previous reports (11, 32). NT, not tested.

5. *I think the classification of endo-acting enzymes is a bit weak when you only have small saccharides as substrates and not longer chains, as cleaving two sugars in from a chain is not obviously exo but also not clearly an activity that cleaves along an extended chain. It would be good to specify this limitation and that tests on longer polysaccharides could provide a more conclusive endo/exo behavior.*

How were enzyme subsites defined? You don't have longer ligands in your structures, so the designation seems somewhat arbitrary. Please clarify this in the text.

→ Thank you for your comments on the classification of endo-type enzymes. We agree with the reviewer that we did not provide sufficient evidence for the endo/exo behavior of the enzymes. As rightly pointed out, experiments using polysaccharides as substrates are essential to conclusively demonstrate the endo/exo behavior of the enzymes; however, such substrates are not available. Therefore, we have opted to designate this enzyme as an oligosaccharide/monosaccharide-releasing (OR/MR-) enzyme, rather than classifying it as an endo/exo-acting enzyme. (please see our general,

opening response above).

As the subsite position was estimated based on the position and size of the sugar moiety in the docking model with oligosaccharides (Supplementary Fig. 13a, newly added in this revision). The substrate binding site of NgaCa exhibits a cleft structure; in addition, three characteristic tryptophan residues (W197, W218, W249) are located in the subsites -2 and -3 (Supplementary Fig. 13a, which has been newly added in this revision). Such a cleft structure and the presence of aromatic amino acids (such as tryptophan or phenylalanine) in the substrate binding site are characteristically observed in endo-type enzymes, such as chitinase or xyloglucanase, which bind long oligosaccharides (references below; Kurašin et al., Ariza et al., Matsuzawa et al.). A docking model of NgaCa with oligosaccharide suggested that these tryptophan residues may play a role in a stacking-type interaction with sugars located in subsites -2 and -3. This structural feature indicates that Group 1 enzymes may act on the inner regions of longer sugar chains, in addition to disaccharides from the non-reducing end. These points are elaborated in the Result section (Pages 11, Lines 309–315).

<References>

[Kurašin et al. described that “The long chitin binding cleft is lined with aromatic residues, mostly Trp and Phe (Fig. 1). The hydrophobic interactions form a flexible sheath necessary for substrate recognition and sliding of the polymer chain between successive glycosidic bond cleavages.” (JBC, 290, 29074-85, 2015)].

[Ariza et al. described that “Substrate recognition is achieved by the interactions between aromatic residues and pyranose rings and several hydrogen bonds with hydroxyl groups of pyranose rings. Around the catalytic center, subsites -1, -2, and -3 form hydrogen bonds, and subsites -1, -3, and -4 form aromatic stacking with substrates.” (JBC, 286, 33890-33990, 2011)]

[Matsuzawa et al. described that “*Paenibacillus* endo-xyloglucanase has four characteristic tryptophan residues (W61, W64, W318, and W319) around the active site cleft. W61 and W64 are found in the “negative” subsites. In contrast, W318 and W319 are found in the “positive” subsites. Exo-type enzymes, such as *Geotrichum* OXG-RCBH and *Aspergillus* OREX, lack tryptophan residues analogous to W61, W64, W318, and W319”. (FEBS Letters, 588, 1731-38, 2014)].

6. *I also think the authors should explain why they would look for GalNAc-active enzymes in the deep sea. The rationale for the study is quite vaguely described, for instance on lines 67-70, a need to find new β -NAGs based on little being known currently is in my opinion only a starting point to the aim of the study – why do we need more information? What is, or could be, their relevance? More concrete*

examples of GalNAc containing glycans would be helpful to put the study into a broader context, and I think showing these in a figure would be helpful for the reader, which could be done with symbol nomenclature or actual chemical structures. In connection, the reason to look for these enzymes in the deep sea is somewhat unclear. Why was this environment chosen? Clearly you discovered terrestrial enzymes afterwards through bioinformatic means, so what could be the roles of the enzymes in a deep-sea environment compared to in e.g. a plant like Arabidopsis? Stating that “the deep-sea microbiome is an attractive potential bioresource for screening undiscovered enzymes”, lines 370-371, is still vague in terms of the activity you searched for, which clearly was not limited to that environment... I understand if you chose this environment because it was “cool”, but please rephrase it to better frame the story of the manuscript.

→ Thank you for pointing this out. Because β -GalNAc is prevalent in various glycans across the three domains of life (Bacteria, Archaea, and Eukarya) in different ecological niches, β -NGA likely follows a similar distribution pattern. However, β -NGAs have only been identified from four bacterial species of *Bacillota* and *Bacteroidota* in terrestrial soil and human gut environments, and their use has been discussed only in limited environments and species. Thus, we searched for novel β -NGAs using a sequence-based screening approach against environmental metagenomes. Particularly, we focused on the deep-sea environments to investigate the role of β -GalNAc in natural biological processes. The deep sea is characterized by unique features and distinct bacterial flora that are quite different from terrestrial environments. Since β -GalNAc-containing glycans are found in various glycoconjugates, especially in the exopolysaccharides of bacteria and archaea (Supplementary Fig. 1, newly added in this revision), β -NGA is likely to be used to regulate glycans in various exopolysaccharides. Deep-sea metagenomes hold potential for the discovery of novel β -NGA genes, considering the prevalence of β -GalNAc in bacterial and archaeal exopolysaccharides and chondroitin sulfate in marine environments. Although the novel OR- β -NGAs discovered in this study did not directly hydrolyze chondroitin sulfate in our assay system (Supplementary Fig. 7, newly added in this revision), these novel β -NGAs, together with β -glucuronidase and sulfatase, might play a role in the degradation of chondroitin sulfate oligosaccharides, which are typically degraded by chondroitin sulfate lyase. We have added these sentences to the Introduction (Page 4, lines 82–93) and Discussion sections (Page 16, lines 480–483).

7. *The possible biological roles of these enzymes could very well be discussed further, to give some more context as to what they might be active on. For example, on lines 270-271 you state that “This structural feature indicated that Group 1 enzymes act on the inner regions of longer sugar chains”.*

Which types of longer sugar chains are those? From the STRING database analysis, it seems like many of the enzymes could have activities related to exopolysaccharide remodeling, since you have a mixture of transporters, degradative, and synthesizing enzymes, which could be expanded by giving some more examples of how these structures may look. In connection to this, I think it would be helpful to show the structures of the substrates tested, either in the main text or in the supplementary information (either using symbol nomenclature or actual chemical structures), as the extended data will not be as accessible to the reader.

→ Thank you for your very helpful comments. As you have pointed out, it is assumed that the Group 1 enzyme degrades some long sugar chains, such as exopolysaccharides, capsular polysaccharides, and the *O*-antigen. The revised manuscript now includes a discussion suggesting that the newly discovered OR- β -NGA may be involved in the degradation of long repeating glycoconjugates, such as exopolysaccharides, capsular polysaccharides, the *O*-antigen, etc. (Page 15–16, Lines 469–472). In addition, the structure of β -GalNAc-containing glycans has been shown using Symbol nomenclature to visually emphasize that the complex and diverse structures of these polysaccharides (Supplementary Fig. 1, newly added in this revision). Considering the possibility that the actual natural substrates of OR- β -NGA are such long-chain repeating oligosaccharides, the substrate binding site of OR- β -NGA, with a cleft structure and characteristic aromatic amino acids, would serve as a clue to uncovering the biological role of this enzyme. This observation is consistent with the hypothesis that it binds long-chain oligosaccharides.

Once again, we thank you for the thoughtful suggestions and insights, which have enriched the manuscript and produced a better and more balanced account of the research.

REVIEWERS' COMMENTS

Reviewer #1 (Remarks to the Author):

Accept manuscript. I am happy with the revisions.

Reviewer #2 (Remarks to the Author):

The authors have taken into account all my comments and have satisfyingly responded to the suggestions. I have no further comments to add, this is a thoroughly conducted piece of work.

Reviewer #3 (Remarks to the Author):

I would like to thank the authors for the very detailed and comprehensive revision. I believe they have addressed almost all of my previous concerns in a satisfactory manner.

The only objection I still have is that I don't think any of the enzymes were actually saturated properly. The rule of thumb is to measure the activity at concentrations of $\sim 10 \times K_m$, and only for NgaP2 there is an actual beginning of a plateau. Your limitation of solubility is an ok reason to not go further, but it should be clearly stated that this prevented actual saturation. I think even for NgaP2 it is borderline, but for sure the other reactions cannot be said to reach saturation.

We would like to extend our gratitude to the reviewers for thoroughly evaluating our manuscript.

Below, we have provided our point-by-point responses to the feedback provided by reviewer #3.

Response to Reviewer #3's comments

I would like to thank the authors for the very detailed and comprehensive revision. I believe they have addressed almost all of my previous concerns in a satisfactory manner.

The only objection I still have is that I don't think any of the enzymes were actually saturated properly. The rule of thumb is to measure the activity at concentrations of $\sim 10 \times K_m$, and only for NgaP2 there is an actual beginning of a plateau. Your limitation of solubility is an ok reason to not go further, but it should be clearly stated that this prevented actual saturation. I think even for NgaP2 it is borderline, but for sure the other reactions cannot be said to reach saturation.

→ Thank you for your comprehensive explanation. We have revised the manuscript per your suggestions.

Kinetic assays were performed at a maximum substrate concentration of 2 mM as GalNAc- β -*p*NP was insoluble at buffer concentrations exceeding 3 mM. Nevertheless, all the assayed enzymes did not attain saturation under the aforementioned conditions (Supplementary Fig. 5). Therefore, these enzymes exhibited low affinities for *p*NP-substrates; these kinetic quantities were deemed apparent values owing to the failure to accomplish saturation. Table 1 presents the aforementioned apparent values within parentheses.